# Early Stroke Induces Long-Term Impairment of Adult Neurogenesis Accompanied by Hippocampal-Mediated Cognitive Decline

**DOI:** 10.3390/cells8121654

**Published:** 2019-12-17

**Authors:** Carolin Kathner-Schaffert, Lina Karapetow, Madlen Günther, Max Rudolph, Mahmoud Dahab, Eileen Baum, Thomas Lehmann, Otto W. Witte, Christoph Redecker, Christian W. Schmeer, Silke Keiner

**Affiliations:** 1Hans-Berger Department of Neurology, Jena University Hospital, Am Klinikum 1, 07747 Jena, Germany; Carolin.Kathner-Schaffert@med.uni-jena.de (C.K.-S.); Lina.Karapetow@med.uni-jena.de (L.K.); Madlen.Guenther@med.uni-jena.de (M.G.); Max.Rudolph@uni-jena.de (M.R.); Mahmoud.Mohamed@med.uni-jena.de (M.D.); Eileen.Baum@noldus.com (E.B.); Otto.Witte@med.uni-jena.de (O.W.W.); Christoph.Redecker@klinikum-lippe.de (C.R.); Christian.Schmeer@med.uni-jena.de (C.W.S.); 2Institute of Medical Statistics and Computer Science, University Hospital Jena, Friedrich Schiller University Jena, 07743 Jena, Germany; Thomas.Lehmann@med.uni-jena.de

**Keywords:** aging, cognitive deficits, dentate gyrus, ischemia, search strategies

## Abstract

Stroke increases neurogenesis in the adult dentate gyrus in the short term, however, long-term effects at the cellular and functional level are poorly understood. Here we evaluated the impact of an early stroke lesion on neurogenesis and cognitive function of the aging brain. We hypothesized that a stroke disturbs dentate neurogenesis during aging correlate with impaired flexible learning. To address this issue a stroke was induced in 3-month-old C57Bl/6 mice by a middle cerebral artery occlusion (MCAO). To verify long-term changes of adult neurogenesis the thymidine analogue BrdU (5-Bromo-2′-deoxyuridine) was administrated at different time points during aging. One and half months after BrdU injections learning and memory performance were assessed with a modified version of the Morris water maze (MWM) that includes the re-learning paradigm, as well as hippocampus-dependent and -independent search strategies. After MWM performance mice were transcardially perfused. To further evaluate in detail the stroke-mediated changes on stem- and progenitor cells as well as endogenous proliferation nestin-green-fluorescent protein (GFP) mice were used. Adult nestin-GFP mice received a retroviral vector injection in the hippocampus to evaluate changes in the neuronal morphology. At an age of 20 month the nestin-GFP mice were transcardially perfused after MWM performance and BrdU application 1.5 months later. The early stroke lesion significantly decreased neurogenesis in 7.5- and 9-month-old animals and also endogenous proliferation in the latter group. Furthermore, immature doublecortin (DCX)-positive neurons were reduced in 20-month-old nestin-GFP mice after lesion. All MCAO groups showed an impaired performance in the MWM and mostly relied on hippocampal-independent search strategies. These findings indicate that an early ischemic insult leads to a dramatical decline of neurogenesis during aging that correlates with a premature development of hippocampal-dependent deficits. Our study supports the notion that an early stroke might lead to long-term cognitive deficits as observed in human patients after lesion.

## 1. Introduction

Ischemic stroke is a major cause of long-term disability and death worldwide. Recover from initial paresis, movement problems, and sensory disturbance and/or aphasia is often associated with cognitive impairments resulting from the stroke [1,2]. Moreover, progressive decline in cognitive function after an ischemic stroke in the subcortical or cortical areas of the brain doubles the risk of dementia. In addition, deficits in targeted attention as well as visual-spatial performances and depression were reported after a stroke [1]. Most of affected individuals did not show any improvement in cognitive functions two years after the lesion [2]. Furthermore, 50% still showed a below-average performance even ten years following injury [3]. Aside from vascular dementia, the causes of these cognitive impairments have not yet been sufficiently clarified, not the least because of their diversity. However, it is noticeable that patients with cortical or subcortical infarcts have impaired cognitive functions [4]. Cognitive deficits associated with cortical or sub-cortical areas result either directly from dysfunction of the affected area or from hypoperfusion in adjacent tissues, as well as from a dysfunction in remote brain areas including the hippocampal formation. This brain region is deeply involved in learning and memory consolidation. It has been suggested that cortical and subcortical infarcts disturb the integrity of the complex hippocampal network essential for proper function and thereby contributes to cognitive impairment [5,6]. However, the underlying mechanisms responsible for cognitive dysfunction are still poorly understood.

Numerous studies indicate that new neurons are generated throughout a lifetime in the dentate gyrus region of the hippocampal formation in the healthy brain [7,8,9]. Following a stroke, neural progenitor cells increase their proliferation rate leading to the formation of new neurons. Newly formed neurons functionally integrate into the existing network and contribute to learning and memory. However, following lesion, morphologically aberrant neurons also appear in addition to the regularly integrating neurons [5,6]. These aberrant neurons are characterized by bipolar dendritic arborization and ectopic location, and are also able to integrate into the hippocampal network [5,6]. Moreover, aberrant neurogenesis has been associated with hippocampal-dependent memory deficits [6].

Most studies evaluate the impact of stroke on brain function in the short-term following lesion [10,11,12]; however, to what extent a stroke lesion early in life affects neural precursor populations, neurogenesis and integration of new neurons over an extended period of time after lesion, is still not fully investigated. Furthermore, whether alterations in the neurogenic niche are associated with changes in brain function (i.e., learning and memory) has yet to be addressed. Therefore, here we evaluated the impact of a prefrontal stroke induced in the young mice on the neurogenic niche and cognitive function in mice during aging. We hypothesize that a stroke lesion in young mice, which significantly increases neurogenesis during the first weeks, will disturb the neurogenic niche, and lead to a long-lasting cognitive impairment during aging. To test our hypothesis, we induced a stroke in 3-month-old mice using the middle cerebral artery occlusion (MCAO) model and evaluated the cellular and functional consequences in the dentate gyrus during aging. To assess long-term effects on cognitive outcome we used a modified version of the Morris water maze, which permits the use of a re-learning paradigm and the differentiation between hippocampus dependent- and independent search strategies.

Our study clearly demonstrates that MCAO in young adult mice leads to a significant reduction of dentate neurogenesis and disturbs endogenous proliferation over the lifespan. These changes in the neurogenic niche are correlated with impairments in flexible learning and deficits in the usage of hippocampal-dependent strategies.

## 2. Materials and Methods

### 2.1. Animals and Experimental Design

The study was performed with a total of 48 male C57Bl/6J mice (3-month-old) and 27 nestin- green-fluorescent protein (GFP) mice (4–6 months old; Figure 1). The nestin-GFP mice were used to evaluate the different precursor subpopulations. The C57Bl/6J mice were randomly divided in three groups: group 1 (6-month-old mice; MCAO n = 8; Sham n = 8), group 2 (7.5-month-old mice; MCAO n = 7; Sham n = 9), and group 3 (9-month-old mice; MCAO n = 8; Sham n = 8; Figure 1A). Nestin-GFP mice (group 4) were divided into MCAO (n = 16) and Sham (n = 11), and received a retroviral vector injection 4 days after surgery (Figure 1B). Cognitive function was assessed in C57Bl/6J and nestin-GFP mice using the Morris water maze (MWM) test for 5 days, one week (group 1–3) or 7 weeks (group 4) before perfusion. For nestin-GFP mice, the MWM was performed before BrdU injections in order to avoid an influence on stem- and precursor cell proliferation (Figure 1B). The C57Bl/6J mice were transcardially perfused at 3, 4.5, and 6 months and nestin-GFP mice at 14–16 months after infarct induction (Figure 1). Mice were held in standard cages (54 cm × 38 cm × 19 cm) on a 14 h light/10 h dark light circle and food pellets and water ad libitum. All procedures were approved by the German Animal Care and Use Committee in accordance with European Directives.

### 2.2. Induction of Brain Infarcts

Ischemic infarcts were induced in 3 months (group 1–3) and 4–6 months (group 4) old mice using the middle cerebral artery occlusion model (MCAO). Mice were anesthetized with 2.5% isoflurane in a N_2_O:O_2_ (3:1) mixture. A midline-neck incision was performed to expose the right common carotid artery (CCA). The two bifurcation of the CCA, the external carotid artery (ECA) and the internal carotid artery (ICA), were localized and cleaned from surrounding tissue. The CCA and ECA were closed with a 7.0 polyfilament (Medicon eG, Tuttlingen, Germany). A 6.0 monofilament suture (Doccol Corporation, Sharon, MA, USA) with a rounded tip was inserted into the ICA. The procedure leads to the occlusion of the middle cerebral artery. The MCAO was performed for 45 min in the C57Bl/6J mice and for 30 min in the nestin-GFP mice, in order to keep the lesion volume homogenous between the mice strains used in the study. The suture was then removed and the wound closed. During MCA occlusion, mice body temperature was maintained using a heating pad. Sham-operated control mice underwent the same surgical procedure except for filament occlusion [6,13].

### 2.3. Injection of Retroviral Vector and BrdU

A red fluorescent protein (RFP)-retroviral vector was injected into the dentate gyrus of the 20-month-old group on day 4 after surgery, in order to determine the impact of stroke on the morphology of newly generated neurons, according to a procedure previously described [5,6]. The CAG-red fluorescent protein (RFP) retroviral vectors were developed from a mouse Moloney leukemia virus by co-transfection of HEK 293 T cells with the compound promoter CAG, the reporter gene RFP, the CMV enhancer protein, the VSV-G rabies virus coating glycoprotein, and the woodchuck hepatitis virus post-transcriptional regulatory element (WPRE). The final titer was 1 × 10^7^ colony forming units/mL. For the injections, mice were anesthetized with 2.5% isoflurane in a N_2_O:O_2_ (3:1) mixture. A sagittal section was made to open the scalp. The following coordinates were used: lateral −1.5 mm from the midline and −1.9 mm posterior to bregma. A glass cannula, containing 1.2 µL of viral vector was inserted into the opening from the scalp, dorsoventral from the dura mater, 2 mm deep into the brain tissue on the side of the stroke. During injection, body temperature was maintained using a heating pad. Sham-operated control mice underwent the same surgical procedure [6].

For labeling of proliferating cells, animals were treated with 5-bromo-deoxyuridine (BrdU; 50 mg/kg, i.p.) twice daily for five consecutive days, six weeks before perfusion (Figure 1).

### 2.4. Morris Water Maze (MWM)

In order to assess cognitive function, a Morris water maze test was carried out [14]. In this test, mice have to find a hidden platform (d = 18 cm; 1 cm beneath the water surface) in a large water pool (diameter: 180 cm). The pool was filled with milky water at a temperature of 20 ± 1 °C. The experiment was divided into two phases: an acquisition and a reversal phase. For the acquisition phase, the platform was positioned in the middle of the northeast quadrant on days 1–3 and for the reversal phase on days 4 and 5, it was placed on the opposite side in the southwest quadrant. On each experimental day, mice had to find the platform from another starting position. Each mouse trial was run six times per day and each trial lasted either as long as it took the mouse to find the platform or for 120 s. After 120 s, mice were guided to the platform, where they waited for 15 s before being returned to the cage. Each trial was tracked with a video camera using the Videomod 2 (TSE version 6.04; Germany) software. A probe trial was performed at the beginning of day 4 and at the end of day 5 [6]. For the probe trial, the platform was removed and each trial lasted 60 s. Learning performance was evaluated using the parameters latency, distance, and velocity to reach the platform, and the search strategies used.

The search strategies were classified into hippocampus-dependent and independent. The hippocampus-dependent strategies include: direct search (strat7), characterized by a constant course in the direction of the platform (>80% of the time in the target corridor); focal search (strat6), characterized by a highly localized search near the platform (≥50% of the trial in the target zone); directed search (strat5) characterized by the preference for a corridor towards the platform or the platform quadrant (>80% time in target corridor); chaining (strat4), defined by the search close to the correct radial distance from the platform to the wall; scanning (strat3), defined by a preference for the central pool area; thigmotaxis (strat1), characterized by maintaining proximity to the wall (>65% of the time in wider wall zone and >35% in closer wall zone); perseverance (strat8), characterized by an erroneous preference for a non-target area (>60% attempt in one or >75% in two adjacent non-target quadrants); and random search (strat2), defined by >60% surface coverage (Figure 9). Analyses of the different search strategies were performed on video recordings from each animal and trial using the Matlab software and an algorithm previously described [14]. The algorithm evaluates the swimming path from the animals in each trial and assigns the corresponding strategy according to the criteria previously described. The percentage of usage of each strategy per day for every group is then calculated. In addition, the hippocampus-dependent strategies were analyzed by determining the percentage of direct swim, focal, and directed search from each animal per trial per day. The mean of the percentage of hippocampus-dependent strategies from stroke versus sham groups per day for each age group is depicted in Figure 9.

### 2.5. Tissue Preparation and Immunocytochemistry

Mice were anesthetized and transcardially perfused with 4% paraformaldehyde in 0.1 M phosphate buffer. Brains were removed, postfixed for 24 h and cryoprotected with 10% and 30% sucrose. Brains were sliced into 40 µm sections using a freezing microtome and stored at −20 °C [15]. Peroxidase staining was used to determine the volume from infarct area, whole brain, hippocampus, and dentate gyrus. Ki67 and BrdU staining were used for quantification of cell proliferation, differentiation, and survival.

Immunocytochemistry was performed on free floating sections. Sections were incubated for 30 min with 0.6% hydrogen peroxide in tris-buffered saline (TBS), washed several times with TBS, denaturized with 2 N hydrochloric acid for 30 min and rinsed with 0.1 M boric acid for 10 min. After washing several times with TBS and blocking with 3% donkey serum and 0.1% triton in TBS (TBS-plus solution) for 30 min, slices were incubated in primary antibody in TBS-plus (Appendix A) overnight at 4 °C. After washing with TBS and blocking with TBS-plus solution, sections were incubated with secondary antibodies diluted in TBS-plus (Appendix A) for 2 h at room temperature. Sections were rinsed and incubated in avidin–biotin–peroxidase complex (Vector Laboratories, Burlingame, CA, USA) for 60 min. Labeled cells were visualized using 3.3-diaminobenzidine solution (0.25 mg/mL, Sigma Aldrich, Munich, Germany). Sections were washed, mounted, and covered.

Immunofluorescence on free floating sections was used to identify BrdU-positive cells. Slices were denatured with 2 N hydrochloric acid for 30 min and rinsed with 0.1 M boric acid previously described. Sections were washed with TBS and blocked in TBS-plus solution for 30 min followed by incubation with primary antibodies (Appendix A) in TBS-plus overnight at 4 °C. After washing with TBS and blocking with TBS-plus solution, sections were incubated in secondary antibody diluted in TBS-plus (Appendix A) for 24 h at 4 °C. After washing several times with TBS sections were mounted with Moviol (Calbiochem, Frankfurt, Hessen, Germany).

Precursor cells subpopulations were identified immunocytochemically using triple-labeling and RFP-positive neurons by double-labeling. After washing steps and blocking with TBS-plus solution (6% serum), sections were incubated with a primary antibody cocktail for 24 h at 4 °C. After washing and blocking for 30 min with TBS-plus solution, sections were incubated with secondary antibodies (Appendix A) for 2 h at room temperature. After rising with TBS, slides were stained with DAPI solution (Sigma Aldrich, St. Luis, MO, USA) in a cuvette for 5 min, washed in phosphate-buffered saline and mounted with Moviol.

### 2.6. Analysis of the Dendritic Arborization in Virally-Labeled Neurons

The complexity of dendritic trees in RFP viral-labeled neurons was quantitated by means of a sholl analysis. Sections of 40 μm in thickness were prepared and the RFP signals were amplified using immunofluorescence. Z-stacks of RFP-positive cells were performed in the dentate gyrus and a Sholl analysis was carried out by using ImageJ with the “Sholl analysis” plugin. The interval between concentric circles was 5 μm with the center point at the soma.

### 2.7. Volume Assessment and Cell Number Quantifications

For the analysis of the infarct area, whole brain, hippocampus, and dentate gyrus volume, each sixth MAP2 peroxidase stained section (from Bregma 3.20 mm/Interaural 7.00 mm to Bregma −4.70 mm/Interaural −0.92 mm) was assessed with a digital camera (Hamamatsu Photonics K.K., San Jose, CA, USA) and measured using Simple PCI software (version 6, Hamamatsu Photonics K.K, San Jose, CA, USA). The area of each section was measured and the value was multiplied by the section thickness (40 µm) as previously described [6,16].

Quantification of BrdU- and Ki67-positive cells was performed employing an Axioplan 2 imaging microscope (Carl Zeiss, Jena, Germany). The number of positive cells on ipsi- and contralateral sides of the brain was determined in every sixth section.

To quantify the distinct cell subpopulations and PCNA-positive cells in the SGZ, DAPI stained nuclei in every 12th section were co-labeled for both nestin-GFP and glial fibrillary acidic protein (GFAP; type 1), only for nestin-GFP (type 2a), for nestin-GFP and doublecortin (DCX; type 2b), or only for DCX (type 3, with short horizontal processes) and assessed using confocal microscopy (LSM 710, Carl Zeiss Jena, Germany). The total numbers of each cell subpopulation were calculated by multiplying the cell counts with the factor 12 [6,12].

Quantification of new neurons was undertaken for every twelfth section (480 μm interval) along the dentate gyrus for both hippocampi by analyzing the co-expression of BrdU and NeuN. The total number of newly generated neurons was calculated by multiplying the percentage of BrdU-positive cells co-expressing the neuronal marker NeuN with the corresponding total number of BrdU-positive cells in the dentate gyrus, as previously described [17].

### 2.8. Statistical Analysis

All data for cell quantifications, volumetry and probe trail of the MWM were calculated by a Mann–Whitney U test due to their skewed distribution. Median (Mdn) as well as interquartile range (IQR) are reported for each group. In this exploratory analysis, each *p* value should be considered as the level of evidence against each null hypothesis. Therefore, nominal *p* values without adjustment for multiple testing are presented.

Sholl analysis of dendritic complexity in the 20-month-old group was tested using the one-way ANOVA (dependent variable: intersections, factor: groups sham versus MCAO). The data are reported as mean ± SEM.

For classic parameters (latency, distance and velocity) and hippocampus-dependent and -independent strategies of the water maze, statistical analyses were performed as follows:For the analyses of latency, distance or velocity in both groups (MCAO, sham control) at the different ages, a 2 way-ANOVA with repeated measures and post-hoc Tukey test was performed (dependent variable: latency, distance or velocity; inner-subject variables: days and trails; between subject factor: ages).For the analysis of latency, distance or velocity in both groups (MCAO, sham control) at the different ages, a 2 way-ANOVA with repeated measures and post-hoc Tukey test was used (dependent variable: latency, distance or velocity; inner-subject variables: days and trails; between subject factor: groups).For the comparison of latency, distance or velocity in both groups (MCAO versus sham controls) on each day at the different ages, a 2 way-ANOVA with repeated measures and post-hoc Bonferroni test was used (dependent variable: latency, distance or velocity; inner-subject variables: trails; between subject factor: groups).For the analysis of hippocampus-dependent strategies in both groups (MCAO, sham controls) at the different ages, groups and days a binary logistic regression and post-hoc Bonferroni test was performed (dependent variable: hippocampus-dependent strategies; subject variables: animal; between subject factor: ages, days, groups or interaction between days and groups).In order to characterize the use of hippocampus-dependent strategies by both groups (MCAO, sham controls), at the different days and ages, we performed an exploratory data analysis by applying a binary logistic regression (dependent variable: hippocampus-dependent strategies; subject variables: animal; between subject factor: groups).For the analysis of each strategy in both groups (MCAO, sham controls) at the different ages, groups and days a binary logistic regression and post-hoc Bonferroni test was performed (dependent variable: each strategies (strat1 to strat8); subject variables: animal; between subject factor: ages, days, groups, or interaction between days and groups).The different hippocampal-dependent and -independent search strategies used in the MWM were analyzed performing an exploratory data analysis by means of an algorithm based on the generalized estimating equations method [14].

Classical parameters of the MWM (latency, distance, and velocity) and hippocampus-dependent strategies are given as mean ± SEM.

Statistical analyses were performed using SPSS 22.0 for Windows (IBM Corp., Armonk, NY, USA). A *p* value of <0.05 was considered to be statistically significant.

## 3. Results

All stroke animals showed typical subcortical damage in the striatum (Figure 2D). Control mice did not exhibit any structural changes after the sham surgery. Brain volume (Figure 2A); hippocampal volume (Figure 2B); and dentate gyrus volume (Figure 2C) from lesioned and control mice within the 6, 7.5, 9, and 20-month groups showed no statistically significant differences (Figure 2). No statistical differences in brain volume were observed between the MCAO groups (Appendix A). In the sham groups a larger brain volume was found in the 9-month-old group compared to 6-, 7.5-, and 20-month-old groups (Appendix A). The lesion volume was significantly reduced between 6 and 20 months (Figure 2D; Appendix A).

### 3.1. Stroke-Dependent Reduction in Proliferation and Adult Neurogenesis during Aging

An age-dependent reduction in the number of Ki67-positive cells was observed in the dentate gyrus between 6 and 9 months of age in the sham and MCAO groups (sham groups: 6 m versus 9 m: U < 0.001; n = 10; *p* = 0.009; 7.5 m versus 9 m: U < 0.001; n = 10; *p* = 0.009; MCAO groups: 6 m versus 7.5 m: U = 5.00; n = 12; *p* = 0.042; 6 m versus 9 m: U < 0.001; n = 13; *p* = 0.003; 7.5 m versus 9 m: U = 3.00; n = 11; *p* = 0.009; Appendix A).

In addition, a significant stroke-dependent decline was detected in the 9-month-old mice (9-month group: sham Mdn = 300 cells; IqR = 78.00; MCAO Mdn = 132 cells; IqR = 72.00; U < 0.001; n = 11; *p* = 0.006; Figure 3).

Adult neurogenesis significantly decreased during aging and also after MCAO. Neurogenesis was significantly decreased in the 6, 7.5, and 9-month-old group compared to the 20-month-old group (Figure 4, Appendix A).

A significant decrease in adult neurogenesis was also found in 7.5 and 9-month-old mice after lesion, as compared to sham controls (6-month group: sham Mdn = 1614 cells; IqR = 927; MCAO Mdn = 1270 cells; IqR = 924; U = 12.50; n = 12; *p* = 0.432; 7.5-month group: sham Mdn = 1492 cells; IqR = 750; MCAO Mdn = 801 cells; IqR = 374; U = 1.00; n = 12; *p* = 0.005; 9-month group: sham Mdn = 1604 cells; IqR = 680; MCAO Mdn = 1026 cells; IqR = 830; U = 1.00; n = 10; *p* = 0.019; Figure 4A). The 20-month-old group showed no stroke-associated changes in the number of newly formed neurons (sham Mdn = 13 cells; IqR = 28.13; MCAO Mdn = 28 cells; 37.50; U = 15.00; n = 13; *p* = 0.445; Figure 4A). In addition, we found only small differences in the dendritic branching between sham control and MCAO from the 20-month-old group (Figure 4B/Appendix A). At the time point evaluated we did not observe any aberrant neuron.

### 3.2. Stroke-Dependent Changes in the Aged Neurogenic Niche

To further evaluate the stroke-dependent long-term changes in the neurogenic niche, different neural subpopulations were analyzed in the 20-month-old nestin-GFP mice (Figure 5). The distinct precursor cells were quantified by triple-immunofluorescence with antibodies against GFP, GFAP, and DCX. Morphological characteristics and co-localization of specific markers were used to differentiate between five different precursor subtypes as follows: type 1 stem cells that express the astrocytic marker GFAP and nestin-GFP and carry a long apical harbor extending into the molecular layer; type 2 cells that show short horizontally oriented processes and express early neuronal marker (type 2a nestin-GFP; type 2b nestin-GFP, DCX); type 3 cells that express only DCX; the immature neurons that show dendrites and express DCX (Figure 5A).

The largest number of precursors was represented by type 1 stem cells, followed by type 2a cells (sham: 19% ± 9% versus MCAO 20% ± 9%). Type 2b cells represented 6% ± 3% in the sham versus 4% ± 3% in the MCAO group. Type 3 cells represented 6% ± 3% in the sham versus 3% ± 1% in the MCAO group. No stroke-related changes were detected in the type1, type 2, and type 3 cells (Figure 5B).

The 20-month-old sham group showed significantly more immature neurons compared to the infarct animals (type 1 cells: sham Mdn = 1800 cells; IqR = 1404; MCAO Mdn = 2928 cells; IqR = 1728; U = 12; n = 13; *p* = 0.242; type 2a cells: sham Mdn = 624 cells; IqR = 540; MCAO Mdn = 768 cells; IqR = 516; U = 13.00; n = 11; *p* = 0.715; type 2b cells: sham Mdn = 168 cells; IqR = 180; MCAO Mdn = 120 cells; IqR = 162; U = 7.5; n = 11; *p* = 0.169; type 3 cells: sham Mdn = 240 cells; IqR = 228; MCAO Mdn = 132 cells; IqR = 78; U = 11.00; n = 11; *p* = 0.460; immature cells: sham Mdn = 240 cells; IqR = 204; MCAO Mdn = 72 cells; IqR = 90; U = 2.50; n = 11; *p* = 0.022; Figure 5B, Appendix A).

We further analyzed the proliferative response of the distinct subpopulations using the endogenous proliferation marker PCNA (Figure 6). The largest fraction was formed by type 2a cells (sham 51% ± 24% versus MCAO 58% ± 10%). Type 2b cells represented 11% ± 4% in the sham versus 17% ± 13% in the MCAO group. Proliferating type 3 cells were also detected (sham 8% ± 5% versus MCAO 2% ± 4%).

Concerning the absolute cell numbers of each subpopulation, only 1% ± 1% of the type 1 cells in the sham and 2% ± 1% were active in the MCAO group, while other groups showed higher proliferative activity (type 1 cells: sham Mdn = 12 cells; IqR = 48; MCAO Mdn = 42 cells; IqR = 21; U = 7.00; n = 9; *p* = 0.453; type 2a cells: sham Mdn = 192 cells; IqR = 204; MCAO Mdn = 144 cells; IqR = 93; U = 8.00; n = 9; *p* = 0.623; type 2b cells: sham Mdn = 24 cells; IqR = 18; MCAO Mdn = 42 cells; IqR = 75; U = 6.00; n = 9; *p* = 0.317; type 3 cells: sham Mdn = 24 cells; IqR = 18; MCAO Mdn = 0 cells; IqR = 18; U = 5.00; n = 11; *p* = 0.180; Figure 6B). There were no significant differences between the two groups (Appendix A).

We further divided the type 1 stem cells into branched and unbranched types in order to better assess their state of development [18,19] (Figure 7). Unbranched type 1 stem cells were described as having a major apical harbor branching into the molecular layer, whereas the branched type 1 stem cells already branched out in the granular cell layer and had additional basal and somatic spurs (Figure 7A). In both groups, branched type 1 stem cells were significantly higher in the control or MCAO group compared to the unbranched type 1 stem cells (Figure 7B). There were no differences in the distribution of branched and unbranched cells between the stroke and control groups (branched stem cells: sham 65% ± 12%; MCAO Mdn = 69% ± 14%; U = 16.50; n = 13; *p* = 0.607; unbranched stem cells: sham 35% ± 12%; MCAO Mdn = 31% ± 14%; U = 16.50; n = 13; *p* = 0.607; Figure 7B; Appendix A).

### 3.3. Stroke-Dependent Effects on Learning and Memory

In order to assess the learning performance, the classical parameters of latency, swimming distance, and velocity were evaluated (Figure 8).

Stroke and control mice showed typical sigmoidal learning curves between day 1 and day 3 (Figure 8B,C). After changing the platform position on day 4 there was a rise in latency and distance. An improvement in the reversal behavior was discernible on day 5. In sham control mice we found no significant differences in distance between different age-groups but there was a significant difference in latency between 9 and 20 month over 5 days (sham groups distance: *p* = 0.148, F(3) = 1.958; 6 m versus 7.5 m: *p* = 1.000; 6 m versus 9 m: *p* = 0.302; 6 m versus 20 m: *p* = 1.000; 7.5 m versus 9 m: *p* = 0.427; 7.5 m versus 20 m: *p* = 1.000; 9 m versus 20 m: *p* = 0.304; latency: *p* = 0.037; F(3) = 3.337; 6 m versus 7.5 m: *p* = 1.000; 6 m versus 9 m: *p* = 0.237; 6 m versus 20 m: *p* = 1.000; 7.5 m versus 9 m: *p* = 0.371; 7.5 m versus 20 m: *p* = 1.000; 9 m versus 20 m: *p* = 0.031; Appendix A).

We also found no differences in latency between the stroke groups; however, there were significant differences in distance over 5 days. (MCAO groups distance: *p* < 0.001; F(3) = 21.069; 6 m versus 7.5 m: *p* = 1.000; 6 m versus 9 m: *p* < 0.001; 6 m versus 20 m: *p* = 0.001; 7.5 m versus 9 m: *p* < 0.001; 7.5 m versus 20 m: *p* < 0.001; 9 m versus 20 m: *p* = 0.232; latency: *p* = 0.065; F(3) = 2.840; 6 m versus 7.5 m: *p* = 1.000; 6 m versus 9 m: *p* = 1.000; 6 m versus 20 m: *p* = 0.203; 7.5 m versus 9 m: *p* = 1.000; 7.5 m versus 20 m: *p* = 0.131; 9 m versus 20 m: *p* = 1.000). Stroke-dependent changes between 6- and 9-months-old group, 6- and 20-months-old groups, 7.5-month and 9-month-old group, and also between 7.5-month and 20-month-old group were significant.

Analysis of the water maze performance over the 5 days showed significant differences between sham and MCAO in the 6-month-old group (MCAO versus sham groups distance: *p* < 0.001; F(1) = 31.635; 7.5 m *p* < 0.001; F(1) = 39.212; 9m *p* = 0.341; F(1) = 1.024; 20 m *p* = 0.179; F(1) = 2.016 (Figure 8 B/C, Appendix A); latency: 6 m *p* < 0.001; F(1) = 30.755; 7.5 m *p* = 0.004; F(1) = 13.188; 9 m *p* = 0.584; F(1) = 0.326; 20 m *p* = 0.105; F(1) = 3.029.

The comparison between sham and MCAO at the different days revealed significant changes in all groups. During the first 3 days of learning the 6-, 7.5-, and 9-month-old groups showed significant differences, whereas the re-learning of a new platform position was impaired in 7.5-, 9-, and 20-month-old group.

The path length to find the hidden platform was increased in the 6-month-old and 7.5-month-old stroke groups during the first 3 days. During re-learning phase, the 7.5-month-old group and the 9-month-old stroke groups showed higher distances to reach the platform (MCAO versus sham distance: D1: 9 m *p* = 0.018; F(1) = 8.834; D2: 6 m *p* < 0.001; F(1) = 77.833; 7.5 m *p* = 0.042; F(1) = 5.292; D3: 6 m *p* = 0.008; F(1) = 10.246; 7.5 m *p* = 0.046; F(1) = 5.05; D4: 7.5 m *p* < 0.001; F(1) = 29.856; D5: 7.5 m *p* < 0.001; F(1) = 78.973; 9 m *p* = 0.032; F(1) = 6.737; Figure 8B, Appendix A).

Mice in the 6-month-old stroke group showed significantly increased latency on the first and second days compared to the controls. The platform change on day 4 increased the latency significantly in the 7.5-month-old and 20-month-old stroke groups (MCAO versus sham latency: D1 6 m *p* = 0.030; F(1) = 6.174; D2: 6 m *p* < 0.001; F(1) = 60.429; D4: 7.5 m *p* = 0.004; F(1) = 13.639; D5: 7.5 m *p* < 0.001; F(1) = 38.874; 20 m *p* = 0.042; F(1) = 4.816; Figure 8C, Appendix A).

The stroke animals in the 7.5-month-old group showed significantly higher latency and distance during the reversal on the fourth and fifth days in contrast to the controls. The 6-month-old stroke group showed strong deficits in the first learning phase. In the 7.5-month-old stroke group the learning and re-learning was affected, whereas the 9-month-old and 20-month-old-group showed deficits in re-learning. This impairment in re-learning in the 7.5-, 9-, and 20-month-old groups correlated with a significant decrease in neurogenesis.

The analysis of the velocity indicates a general decrease in the 9-month-old groups for the sham as well as for the MCAO group. The MCAO groups tended to show a higher velocity than the sham groups in the 7.5-month-old group. Thus, an impairment in velocity due to the stroke, as compared to the sham controls, could not be recorded (Figure 8D, Appendix A).

The comparison between MCAO and sham at different days after lesion showed significant differences on days 2, 3, and 4 in the 7.5-month-old group, on day 1 in the 9-month-old and day 1 and day 5 in the 20-month-old group (MCAO versus sham D1: 9 m *p* = 0.019; F(1) = 8.605; 20 m *p* = 0.008; F(1) = 8.948; D2: 7.5 m *p* = 0.041; F(1) = 5.36; D3: 7.5 m *p* = 0.001; F(1) = 38.767; D4: 7.5 m *p* = 0.011; F(1) = 9.224; D5: 20 m *p* = 0.039; F(1) = 4.97; Figure 8D; Appendix A).

Evaluation of the probe trails took place on days 4 (before reversal) and 5 (after reversal) (Figure 9). The probe trails were used to evaluate the preference towards the learned platform position. The 6- and 7.5-month-old mice showed no differences in the probe trial on day 4, whereas the 9-month-old stroke mice showed significantly decreased learning performance compared to sham controls. The 20-month-old stroke group showed no significant differences to the controls. (NE-quadrant: 6 month group: sham Mdn = 54%; MCAO Mdn = 35%; U = 17.00; n = 14; *p* = 0.366; 7.5 month group: sham Mdn = 58%; MCAO Mdn = 45%; U = 13.00; n = 14; *p* = 0.156; 9 month group: sham Mdn = 62%; MCAO Mdn = 36%; U = 2.00; n = 11; *p* = 0.017; 20 month group: sham Mdn = 32%; MCAO Mdn = 46%; U = 49.00; n = 23; *p* = 0.321; Figure 8). The probe trail on day 5 showed significantly reduced preference for the new platform position in the 7.5- and 9-month-old stroke group (SW-quadrant: 6 month group: sham Mdn = 46%; MCAO Mdn = 51%; U = 16.00; n = 12; *p* = 1.00; 7.5 month group: sham Mdn = 57%; MCAO Mdn = 33%; U = 3.00; n = 14; *p* = 0.005; 9 month group: sham Mdn = 56%; MCAO Mdn = 38%; U = 1.00; n = 11; *p* = 0.017; 20 month group: sham Mdn = 35%; MCAO Mdn = 38%; U = 50.00; n = 23; *p* = 0.352; Figure 9, Appendix A).

To conclude, stroke induced in 3-month-old mice showed long-term changes in flexible learning in the MWM, as indicated by altered latencies, distance, and probe trail.

Since neurogenesis is strongly associated with hippocampus-dependent learning, we determined the impact of stroke on strategies used by the animals to find the platform (Figure 10). All groups used hippocampus-independent strategies at the beginning of the test, which changed during ongoing training to more hippocampus-dependent strategies. The ratio of hippocampus-dependent/independent strategies (strat) increased during the water maze performances. After the platform location was changed on day 4, fewer hippocampus-dependent strategies were observed, however, these increased again on day 5 (Figure 10). Whether the strategy development is influenced by age, days and the different groups binary logistic regression analysis with Bonferroni correction were used. The hippocampus-dependent strategies are affected by the different groups, days and group per day (group: *p* < 0.001; df = 1; Chi_Square = 16.605; age: *p* = 0.694; df = 3; Chi_Square = 1.451; day: *p* < 0.001; df = 4; Chi_Square = 110.248; interaction between group and day: *p* = 0.058; df = 4; Chi_Square = 9.144). The detailed analysis of MCAO and sham controls at the different ages reveals less usage of hippocampal-dependent strategies during the learning phase (day 1 to day 3) in the 6- and 7.5-month-old groups and the re-learning phase (day 4 to day 5) in the 6-, 7.5-, and 9-month-old group (MCAO versus sham hippocampal-dependent strategies: 6 m D1 *p* = 0.003, 95% CI (1.828; 16.927); D2 *p* < 0.001, 95% CI (4.336; 208.13); D3 *p* = 0.015, 95% CI (0.13; 22.22); D5 *p* < 0.001, 95% CI (0.19; 12.186); 7.5 m D2 *p* = 0.008, 95% CI (1.02; 9.737); D3 *p* = 0.043, 95% CI (1.042; 5.437); D4 *p* = 0.003, 95% CI (0.039; 0.612); D5 *p* < 0.001, 95% CI (16.03; 775.479); 9 m D5 *p* = 0.016, 95% CI (1.223; 6.823); Figure 10, Appendix A).

For a more detailed analysis, we additionally verified the usage of each hippocampus-dependent and hippocampus-independent search strategy. Global analysis by binary logistic regression analysis with Bonferroni correction revealed dependencies of group, day, age, and interaction between day and group (Figure 10, Appendix A).

Due to the statistical differences, explorative analysis of search strategies for each day and group were verified. During the learning phase, the 6-, 7.5-, and 9-month-old stroke groups used significantly more random search, scanning paths and less direct search paths to locate the platform as compared to sham groups (Figure 10).

After reversal, 6-, 7.5-, and 9-month-old MCAO groups used significantly less hippocampus-dependent strategies on day 5 as compared to sham controls (Figure 10, Appendix A). The 20-month-old stroke group showed significantly more random search paths (Figure 10, Appendix A) compared to the controls at day 4.

Analysis of all three hippocampus-dependent strategies following reversal of the platform revealed that all old sham controls used more specific spatial navigation compared to stroke animals.

Stroke mice showed large and long-term deficits in learning, re-learning, and memory in the Morris water maze, both in terms of classical parameters and hippocampal-dependent strategies.

## 4. Discussion

In the present study, we hypothesized that a stroke induced in adult mice leads to long-term deficits in the neurogenic niche and impairment in learning, re-learning, and memory at higher ages, as described in human stroke patients. To test our hypothesis, we induced a prefrontal stroke lesion in 3-month-old adult mice and evaluated the long-term consequence in 6-, 7.5-, 9-, and 20-month-old mice. Our results clearly demonstrate a stroke-dependent reduction of dentate neurogenesis correlating with deficits in flexible learning and a decline in the usage of hippocampus-dependent strategies, indicating memory impairment.

During adult neurogenesis new neurons are formed, which functionally integrate into the existing neuronal network [20,21,22,23,24]. This is achieved through the proliferation of stem- and progenitor cells and their subsequent differentiation into mature neurons [25,26]. Notably, both age and stroke strongly affect these processes. In particular, previous studies have shown that stem cells, progenitors, and neurogenesis dramatically decrease during the first 6 months of life. After this initial decline, neurogenesis remains stable or decreases slightly until old age. In our study, we focused on the time points where neurogenesis and cognitive function critically change [27,28].

Various stroke models lead to an increase in adult neurogenesis both in young and aged mice on the short-term [29,30]. In order to evaluate the long-term effects of an early stroke lesion, we induced stroke in 3-month old mice and analyzed its effects in 6, 7.5-, and 9-months aged mice. Changes associated with advanced ages were assessed in 20-month-old mice. In comparison to sham mice, there were statistically relevant differences in neurogenesis in lesioned animals between 6-month and 9-month old groups and between 9-month and 20-month old animals. This is in agreement with available evidence indicating that neurogenesis decreases between 9 and 20-month-old mice [28,31]. We also found a statistical decline in neurogenesis in the 7.5 month and 9-month old groups compared to sham controls. This decrease in neurogenesis can be due to the reduction of stem- and progenitor cell populations. Previous studies have shown that aging drives stem cells into quiescence and also differentiation into mature astrocytes, which leads to a loss of stem cells and the capacity to generate progenitors and neurons [18,32,33]. The putative mechanisms leading to increased stem cell quiescence are still under debate.

In contrast to aging, which acts as a negative regulator of neurogenesis, stroke is described as a potentiating stimulus, which dramatically increases stem cell and progenitor proliferation, and dentate neurogenesis directly after infarct induction [34,35,36]. In previous studies, focal infarcts were induced at different ages to analyze the stroke-dependent effects on neurogenesis and cognition [37,38]. Already six hours after a small cortical infarct, proliferation of type 1 stem cells and type 2a cells increases significantly in 3-month old mice. Furthermore, after 24–72 h, type 2b and type 3 cells also show increased proliferation. Following MCAO, the maximum number of proliferating cells was present after six to eleven days [29,39]. Proliferative activity then decreases 2–5 weeks after lesion [39]. The number of neurons rises up 6 weeks after stroke induction. In previous investigations we found that stroke induced in aged mice leads to a dramatic decrease in the proliferation rate of stem and progenitor cells compared to lesioned young mice, however, it increases proliferation compared to sham controls in the same aged group [10]. Here, we evaluated the long-term effects of a stroke induced in young animals on stem- and progenitor cell proliferation and neurogenesis during aging. Both, endogenous proliferation and neurogenesis were significantly diminished below control levels in middle-aged mice. Although stroke stimulates neurogenesis directly after infarct induction, it seems to have an age-associated negative impact on endogenous proliferation and neurogenesis on the long-term, in agreement with previous investigations [6,12]. Therefore, age at stroke onset is a critical determinant of stroke-dependent effects on neurogenesis. Whether stroke-dependent long-term effects on neurogenesis are associated with cognitive impairments, as observed in human patients, is not well understood. In the present study, we focused on the effect of an early stroke on development of cognitive impairment and its correlation to changes in adult neurogenesis in mice. We clearly demonstrated that, following stroke, neurogenesis is reduced below control levels during aging, which is accompanied by a decrease in learning, re-learning, and memory performance. This cognitive impairment might be associated with a decrease in the number of new neurons or a reduced lifespan of newly generated neurons after stroke in older brains. This issue has to be further investigated. Survival of new neurons depends on the integration of these neurons into the existing networks.

The contribution of post-stroke neurogenesis on cognitive function is still under debate. Under physiological conditions, a decrease of neurogenesis due to aging or different interventions (cytostatic drugs and knockout model) correlates with impairments in cognitive function, represented by a reduction of hippocampal-dependent strategies [6,40,41]. Consequently, an impairment in neurogenesis leads to a reduced use of hippocampal-dependent strategies. Under pathophysiological conditions such as a stroke, it is not clear to what extent network activity is disturbed and how these changes are due to the reduction of neurogenesis. Here we show that neurogenesis increased directly after a stroke, which correlated with learning and memory deficits and further with the reduction of hippocampal-dependent strategies in the water maze. Underlying mechanisms were not clear, but might involve neuronal network disturbance caused by the presence of aberrant neurons (already described in young mice after stroke), which leads to the wrong connectivity [42]. A previous study showed that stroke at the age of three months generates aberrant new nerve cells in the dentate gyrus [5]. Moreover, up to 10% of the new neurons exhibited abnormal bipolar morphology with additional basal dendrites in the hilus. In addition, ectopic neurons were detected after incorrect migration into the hilus. In the long run, these aberrant neurons and network connections may also lead to decreased cognition as reflected by a lesser use of hippocampal-dependent strategies. The lifespan of aberrant neurons generated after stroke and their putative contribution to cognitive impairment still needs to be addressed.

On the other hand, it is also feasible that strongly reduced neurogenesis on the long term per se leads to impairment of cognition. The possible mechanisms linking changes in neurogenesis and network connectivity to learning and memory are still not clear and also require further investigations.

The significance of stroke-induced changes in adult neurogenesis for cognitive function of the hippocampus has still not been sufficiently clarified. Under physiological and stroke conditions, new functional neurons integrate into the hippocampal network. Studies have shown a correlation between adult neurogenesis and re-learning for solving tasks in similar spatial environments [43,44,45,46]. The need for new neurons in spatial memory has been shown in studies using various ablation models to suppress neurogenesis [14]. These studies linked reduced neurogenesis to poorer results in spatial learning and memory in the water maze. In particular, the use of different search strategies in the Morris water maze has been described to distinguish between hippocampal- dependent and -independent learning processes in detail. The so-called reversal protocol is considered to be suitable for demonstrating the influence of adult neurogenesis on cognitive flexibility [41]. After the platform is readjusted, a reduction in the number of new neurons results in a much longer-lasting preference for the old target position and the use of independent-hippocampal strategies, mostly in the re-learning phase. Earlier studies showed an age-related impairment of spatial learning in the Morris water maze in rodents, independent of the type of brain lesion [47].

The extent to which stroke-induced neurogenesis contributes to the hippocampal cognitive function is unclear and is to date controversially discussed. Evaluation of the pathway and latency showed a significantly worse re-learning capabilities after the change in the platform position in the stroke groups. In addition, the 6 and 7.5-month-old stroke animals also showed deficits in re-learning with respect to finding the hidden platform position. Further memory deficits were observed in 7.5- and 9-month-old lesion mice. All stroke animals showed less usage of hippocampus-dependent strategies, mainly in the re-learning phase of the task. Previous studies showed that genetic ablation of neurogenesis weakens learning of hippocampus-dependent memory tasks. Blaiss and colleagues [37] used transgenic mice with reduced adult neurogenesis and found no significant differences after a cranial brain injury in fear conditioning and in the rotarod tasks, whereas in the Morris water maze, a worsening in latency and thigmotaxis was observed. Additional studies using cytostatic drugs [48] and irradiation [30] to impair adult neurogenesis and cognitive function, showed a worsening of learning curves compared to controls.

In summary, the present study shows that an early stroke lesion results in a long-term reduction of adult neurogenesis during aging, characterized by both a decline in endogenous proliferation and the number of distinct precursor cell subpopulations. This significant decrease in adult neurogenesis associates with poor flexible learning and memory in the Morris water maze, reflecting deficits in cognitive function of the aging brain.

The present study supports the notion that early stroke lesions might be involved in development of early dementia. Furthermore, results suggest that a lesion-induced decline in adult neurogenesis could be a possible mechanism involved in impaired learning and memory associated with dementia.

## Figures and Tables

**Figure 1 cells-08-01654-f001:**
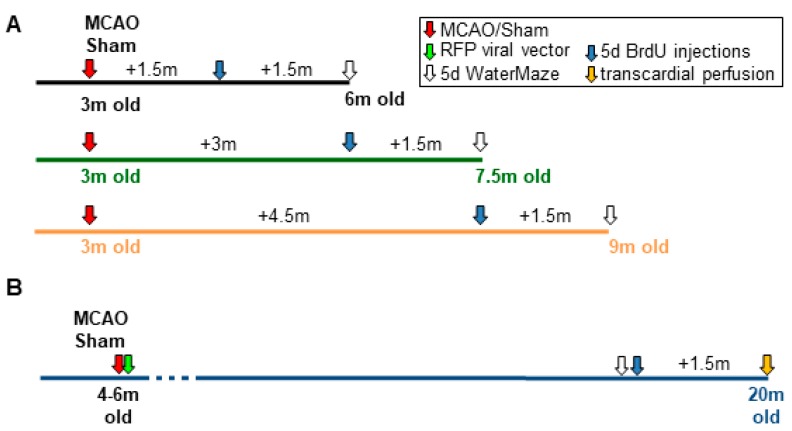
Experimental design. (**A**) After surgery, C57Bl/6J mice received BrdU injections twice daily for five consecutive days at 1.5, 3, and 4.5 month after surgery. The Morris water maze (MWM) was performed 1.5 month after BrdU injections. (**B**) 20-month-old nestin-GFP mice received a retroviral vector injection four days after stroke or sham surgery and MWM was performed 7 weeks before perfusion, followed by an intraperitoneal injection of BrdU twice daily for five consecutive days.

**Figure 2 cells-08-01654-f002:**
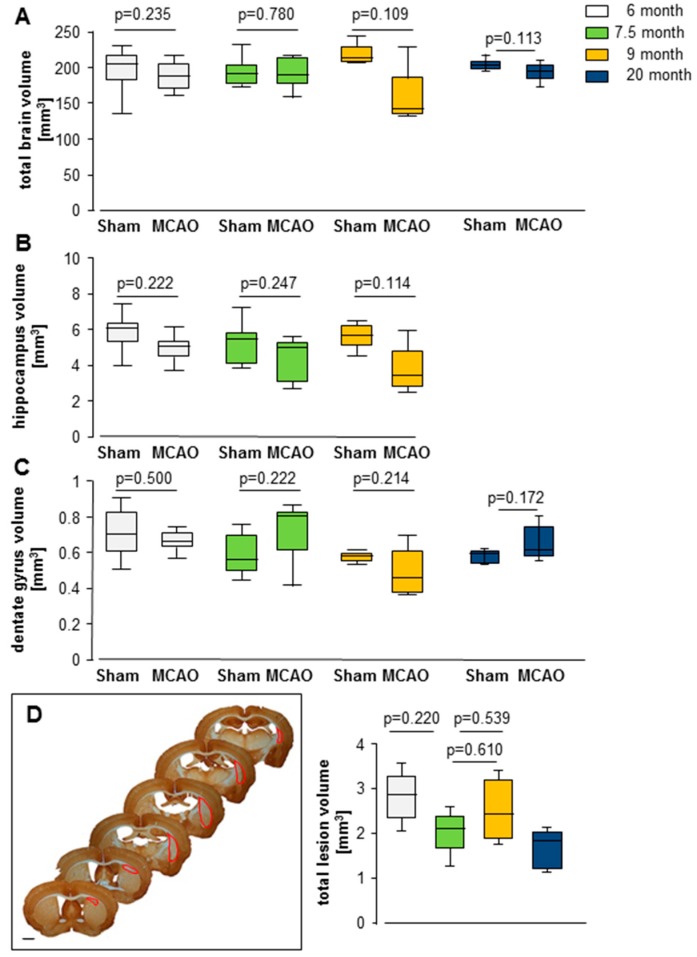
Impact of middle cerebral artery occlusion (MCAO) on (**A**) total brain volume, (**B**) hippocampal volume, (**C**) dentate gyrus volume, and (**D**) total lesion volume. Box plots represent the median, upper and lower quartiles and min and max values. (**D**) Location of the ischemic infarct in MAP2-stained sections and infarct volumetry in the different groups. Red surroundings mark the ischemic infarcts. There are no significant differences between sham and MCAO groups. Analyses were performed using the Mann–Whitney test; all *p*- and n-values are shown in Appendix A; bars = 1 mm.

**Figure 3 cells-08-01654-f003:**
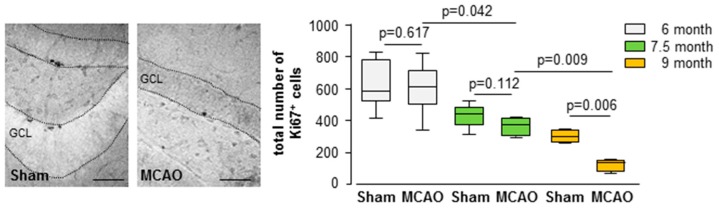
Stroke-induced reduction of endogenous proliferation. Images show Ki67-positive peroxidase-stained section of sham controls and the MCAO group. Box plot represents the median, upper and lower quartiles and min and max values. During aging there is a continuous reduction in the number of proliferating Ki67-positive cells. Analyses were performed using the Mann–Whitney test all *p*- and n-values Appendix A; bars = 100 μm.

**Figure 4 cells-08-01654-f004:**
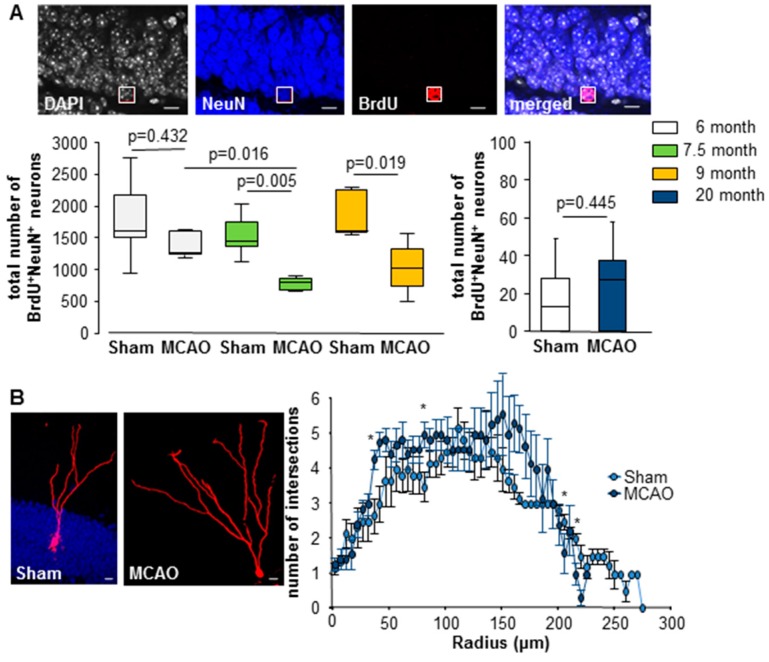
Stroke-induced reduction of newly born neurons. (**A**) Confocal images of immunofluorescent sections for the proliferation marker BrdU (red), mature neurons NeuN (blue), and DAPI (4′,6-diamidino-2-phenylindole, grey) showed newly born neurons. Impaired survival of newly generated neurons in 7.5 and 9-month-old groups. Box plots represent the median, upper and lower quartiles and min and max values. Analysis was performed using the Mann–Whitney test; bars = 10 μm. (**B**) Confocal images of immunofluorescent sections for red-fluorescent protein of the viral vector, bars = 10 μm. Evaluation of the dendritic complexity of virally-labeled neurons using Sholl analysis. The graph represents the means ± SEM using one-way ANOVA, * *p* < 0.05; all *p*- and n-values Appendix A.

**Figure 5 cells-08-01654-f005:**
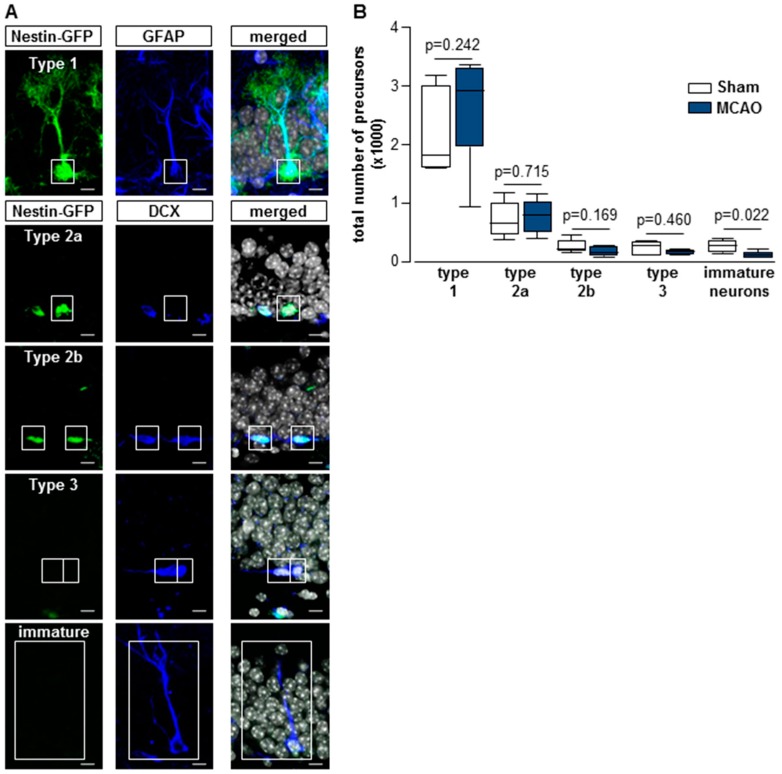
(**A**) Confocal images of immunofluorescent sections for green-fluorescent protein (GFP; green), glial fibrillary acidic protein (GFAP; blue), or doublecortin (DCX; blue) and DAPI (grey) reveal radial glia-like type 1 cells positive for nestin-GFP and GFAP, type 2a cells positive for nestin-GFP but not for GFAP; type 2b cells positive for nestin-GFP and the early neuronal marker DCX; DCX-positive type 3 cells, which lack nestin-GFP and DCX immature neurons containing dendrites, bars = 10 μm. (**B**) Total numbers of precursor cell subpopulations. The number of immature neurons was significantly decreased in the MCAO group as compared to the sham group. Box plots represent the median, upper and lower quartiles and min and max values. Analysis was performed using the Mann–Whitney test; all *p*- and n-values Appendix A.

**Figure 6 cells-08-01654-f006:**
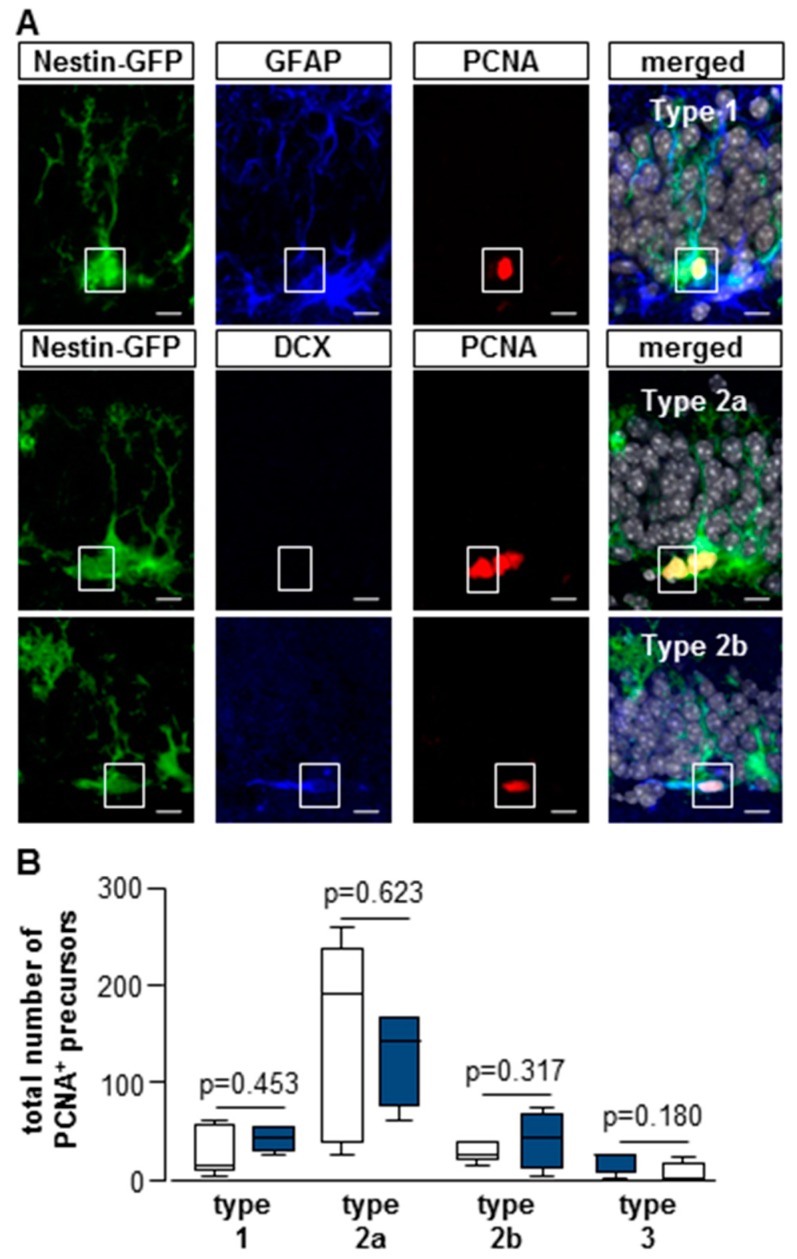
(**A**) Confocal images of immunofluorescent sections for PCNA-positive subpopulations (radial glia-like type 1 cells, type 2a, and 2b cells), bars = 10 μm. (**B**) Total numbers of PCNA-positive precursor cell subpopulations. There were no significant differences in cell proliferation between sham and MCAO groups. Box plots represent the median, upper, and lower quartiles and min and max values. Analysis was performed using the Mann–Whitney test; all *p*- and n-values Appendix A.

**Figure 7 cells-08-01654-f007:**
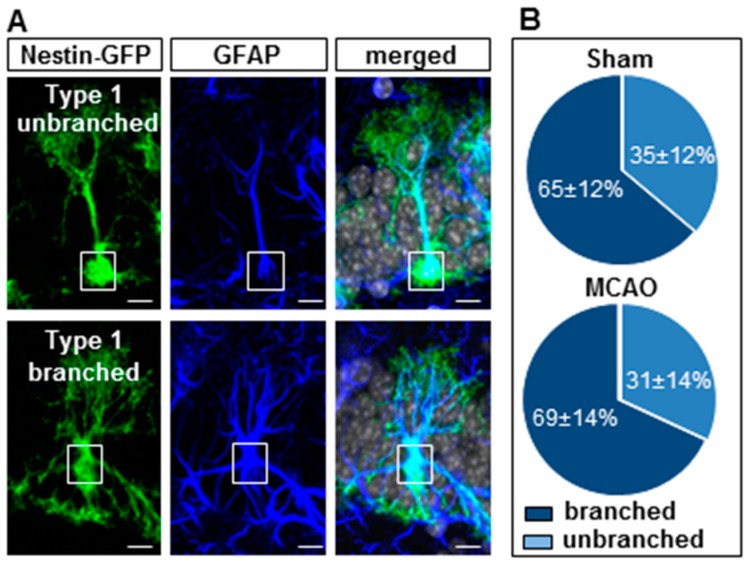
Quantification of branched and unbranched radial glia-like type 1 cells in the 20-month-old nestin-GFP mice. (**A**) Confocal images of immunofluorescent sections for green-fluorescent protein (GFP; green), glial fibrillary acidic protein (GFAP; blue), and DAPI (grey). RGLs, which express nestin-GFP and GFAP, separated into branched and unbranched type 1 cells, bars = 10 μm. (**B**) Stroke did not influence the ratio of branched and unbranched RGLs compared to sham controls. Box plots represent the median, upper and lower quartiles and min and max values. Analysis was performed using the Mann–Whitney test; all *p*- and n-values Appendix A.

**Figure 8 cells-08-01654-f008:**
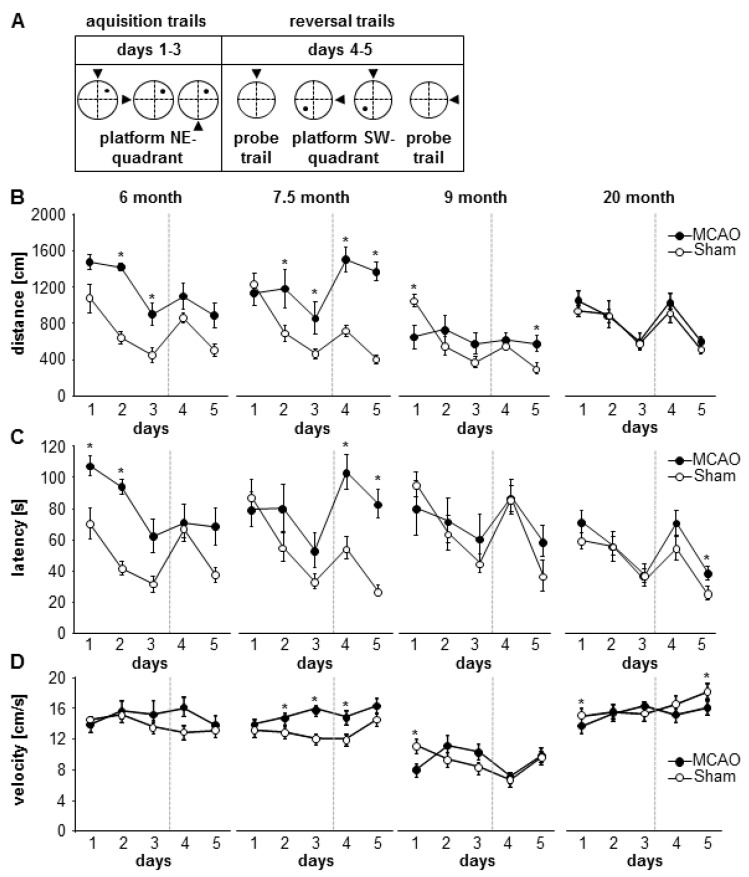
Impact of stroke on learning and re-learning. (**A**) Experimental design of the MWM changing platform position on day 4. Arrows represent the starting position per day. (**B**–**D**) Graphs show distance, latency and velocity to navigate to the platform in MWM. Strong impairments of learning were observed in the 6- and 7.5-month-old mice. Re-learning was reduced in 7.5-, 9-, and 20-month-old MCAO groups. The graph represents the means ± SEM using 2-way ANOVA with repeated measures and post-hoc Tukey (per group) or Bonferroni (per day), * *p* < 0.05; all *p*- and n-values Appendix A.

**Figure 9 cells-08-01654-f009:**
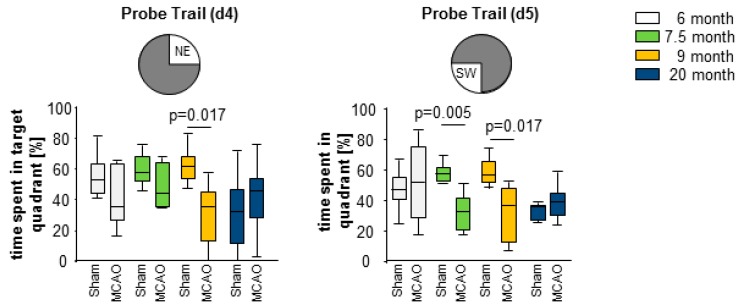
Recall of learning was performed using probe trail on day 4 (before reversal) and day 5 (after reversal). At day 4, most deficits in recall were observed in 9-month-old MCAO mice. At day 5, the 7.5-month-old, and 9-month-old stroke mice showed an impaired spatial learning compared to sham mice in the probe trail. Box plots represent the median, upper and lower quartiles, and min and max values. Analysis was performed using the Mann–Whitney test, all *p*- and n-values Appendix A.

**Figure 10 cells-08-01654-f010:**
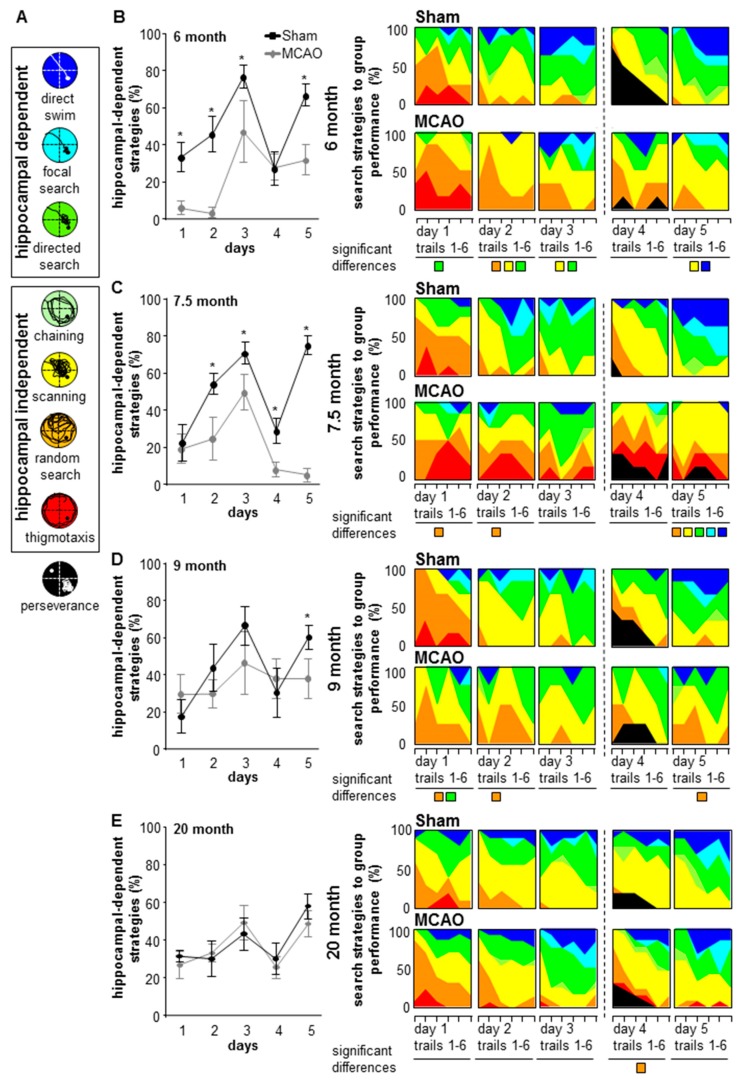
Usage of hippocampal-dependent and independent search strategies in the MWM test. The colorful images show differences in learning and re-learning between MCAO- and sham control mice during aging. (**A**) Each color represents hippocampus-dependent or -independent search strategies. (**B**–**E**) For the analysis of each strategy at the different ages, groups and days a binary logistic regression and post-hoc Bonferroni test was performed. The different hippocampal-dependent and -independent search strategies for each day an exploratory data analysis by means of an algorithm based on the generalized estimating equations method was performed. Strategies with significant differences in usage are represented by colorful squares for each day, * *p* < 0.05. n- and *p*- values Appendix A.

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
