# Peer review of "Early Stroke Induces Long-Term Impairment of Adult Neurogenesis Accompanied by Hippocampal-Mediated Cognitive Decline"

_cells, 2019, doi:10.3390/cells8121654_

Round 1

Reviewer 1 Report

The authors have addressed my questions. I have no more.

Author Response

no comments

Reviewer 2 Report

In this study, the authors found that 60 min-focal ischemia could leads to long-term deficits in the neurogenic niche and impairment in learning, re-learning and memory at higher ages. Although the topic is interesting, there are several concerns regarding the experimental design.

Question is that whether 45 min focal ischemia in mice could cause CA1 region damage. As shown in Figure 1, the infarct is located in striatum, even not in the cortical region. How the hippocampus could be injured? Histological evidence may be provided. If the hippocampus was not damaged in this model, why neurogenesis is affected? The cognitive impairment after sicheic stroke could be due to the white matter injury rather than reduced neurogenesis. BrdU (50 mg/kg, i.p.) was injected twice daily for five consecutive days, six weeks before perfusion. Thousands of new born cells are generated daily, which rapidly migrate to OB or molecular layer of DG. Therefore, these BrdU-positive cells may not truly reflect the proliferating cells in the SGZ. A RFP-retroviral vector was injected into the dentate gyrus, which could lead to a trauma, and in turn effect the neurogenesis. The rational may be provided why water maze test was performed in 6 months after stroke.

Author Response

Reviewer 2

Question is that whether 45 min focal ischemia in mice could cause CA1 region damage. As shown in Figure 1, the infarct is located in striatum, even not in the cortical region. How the hippocampus could be injured? Histological evidence may be provided. If the hippocampus was not damaged in this model, why neurogenesis is affected? The cognitive impairment after ischemic stroke could be due to the white matter injury rather than reduced neurogenesis.

Post-stroke memory dysfunction can be triggered by direct damage to the brain structures involved in memory processing and/or by neurophysiological changes in remote regions that are functionally associated with the ischemic area (diaschisis) (Carrera E. & Tononi G. et al., 2014). It is already clear that the hippocampus, a brain structure with decisive role in the formation of memories, does not operate stand-alone, but embedded in a network of extensive and reciprocal connections to the cingulate cortex and the diencephalon (Bubb E. J. et al. 2017). Several studies clearly show that cognitive deficits imposed by stroke do not simply arise from direct ischemic damage to the hippocampus, but mainly from indirect, delayed distortions of remote, functionally connected regions (Linden J. et al. 2014; Woitke F. et al. 2017; Niv F. et al. 2012). It has been shown that, despite intact anterograde connections from the hippocampus to the thalamus, retrograde connections from the thalamus may be malfunctioning after stroke (Baumgartner et al. 2018). Importantly, ipsilateral thalamic hypometabolism has been found in a rat photothrombosis model of stroke without histological signs of tissue damage in this region (Carmichael, S. T. et al. 2004), indicating that severity, duration or extent of the ischemic damage may determine if remote changes in activity transform into loss of function and cell death. Cell death and structural reorganization in the thalamus after remote stroke have been coined “secondary degeneration”. This phenomenon can arise due to a lack of excitatory input to the thalamus and has been linked to non-specific symptoms after stroke not attributable to the ischemic lesion itself such as cognitive impairment.

On the other hand, there is evidence indicating that brain damage in the mouse occurs in regions including the hippocampus, thalamus and hypothalamus after MCA occlusion (Hata et al., 1998). However, since these areas are not supplied by the MCA, it is unclear how MCAO affects them. It has been proposed that damage induced by proximal occlusion of the MCA in hippocampal, thalamic and hypothalamic areas could be partially related to occlusion of the anterior choroidal artery (AchA, which provides blood supply to the anterior portion of the hippocampus), the lateral hypothalamic artery (LHA) and/or ventral thalamic artery (El Amki et al., 2015). Importantly, several factors affect lesion volume and reproducibility in the MCAO model, including suture coating length and depth of the suture insertion (Shinamura et al., 2006). Our close examination of the hippocampal slices did not reveal obvious structural changes in the hippocampus after MCAO, possibly due to the protocol we used and which was described in previous studies.

Several studies have already shown that, despite an intact hippocampal structure after MCAO, neurogenesis is increased within a short time after lesion. Importantly, there is also evidence indicating that ablation of neurogenesis using irradiation prevents neuronal replacement and functional recovery after MCAO in juvenile mice (Rodgers et al., 2018). Ablation in middle-aged mice recovers motor function after MCAO (Sun et al. 2012) but no cognitive outcome was analyzed. In our study, neurogenesis dropped down significantly during aging after stroke and this was associated with worsening in learning and re-learning. The initial mechanisms driving the disturbance of neurogenesis are completely unclear. There is some evidence suggesting that glia cells and blood vessel can interact with the stem and precursor cells and trigger neurogenesis directly after stroke. As an example, there are several astrocyte signaling pathways that coordinate neural progenitor stimulation, like the Notch pathway. Since astrocytes are interconnected by means of a functional reticular system in the brain termed ‘syncytium’ (Kiyoshi and Zhou, 2019), it is feasible that long-range signals from astrocytes at the lesion site could influence the behavior of these cells in distant regions of the brain and affect homeostasis of several processes including neurogenesis, as part of a systemic response to injury. In support of this hypothesis, findings show that pharmacological blocking of Gap junctional communication significantly increases survival of the pyramidal neurons in the ipsilateral hippocampus and improves behavioral scores in mice after MCAO (Xie et al., 2011). Also, endogenous growth factors such as brain derived neurotrophic factor could be involved. The mechanisms reducing the development of new neurons during aging after stroke are yet completely unknown. Some evidence also suggests that the strong activation of stem- and precursor cells directly after stroke could lead to a consumption of these cells, leading to a decline in neurogenesis. Nevertheless, we found a significant decrease of immature precursor cells in the 20-month-old group.

We agree with the reviewer regarding involvement of ischemic-induced damage to the white matter in cognitive impairment after stroke, as it has already been demonstrated by several studies, however, as pointed out above, there is also direct evidence indicating the role of neurogenesis-associated alterations in cognitive impairment, which was the focus of our study.

Literatur:

Carrera, E. & Tononi, G. (2014) Diaschisis: past, present, future. Brain 137, 2408–2422 Bubb, E. J., Kinnavane, L. & Aggleton, J. P. (2017) Hippocampal - diencephalic - cingulate networks for memory and emotion: An anatomical guide. Brain Neurosci Adv 1 Linden J, Fassotte L, Tirelli E, Plumier JC, Ferrara A. (2014) Assessment of behavioral flexibility after middle cerebral artery occlusion in mice Behav Brain Res. 1;258:127-37 Woitke F, Ceanga M, Rudolph M, Niv F, Witte OW, Redecker C, Kunze A, Keiner S. (2017) Adult hippocampal neurogenesis poststroke: More new granule cells but aberrant morphology and impaired spatial memory, PLoS One. 14;12(9):e0183463 Niv F, Keiner S, Krishna, Witte OW, Lie DC, Redecker C. (2012) Aberrant neurogenesis after stroke: a retroviral cell labeling study, Stroke. 43(9):2468-75 Baumgartner P, El Amki M, Bracko O, Luft AR, Wegener S. (2018) Sensorimotor stroke alters hippocampo-thalamic network activity. Sci Rep. 25;8(1):15770 Carmichael, S. T., Tatsukawa, K., Katsman, D., Tsuyuguchi, N. & Kornblum, H. I. (2004) Evolution of diaschisis in a focal stroke model. Stroke 35, 758–763 Hata R, Mies G, Wiessner C, Fritze K, Hesselbarth D, Brinker G, Hossmann KA. (1998) A reproducible model of middle cerebral artery occlusion in mice: hemodynamic, biochemical, and magnetic resonance imaging. J Cereb Blood Flow Metab. 18(4):367-75. El Amki M, Clavier T, Perzo N, Bernard R, Guichet PO, Castel H. (2015) Hypothalamic, thalamic and hippocampal lesions in the mouse MCAO model: Potential involvement of deep cerebral arteries? J Neurosci Methods. 30;254:80-5 Shimamura N, Matchett G, Tsubokawa T, Ohkuma H, Zhang J. (2006) Comparison of silicon-coated nylon suture to plain nylon suture in the rat middle cerebral artery occlusion model. J Neurosci Methods. 30;156(1-2):161-5 Rodgers KM, Ahrendsen JT, Patsos OP, Strnad FA, Yonchek JC, Traystman RJ, Macklin WB, Herson PS (2018) Endogenous Neuronal Replacement in the Juvenile Brain Following Cerebral Ischemia. 1;380:1-13 Sun F, Wang X, Mao X, Xie L, Jin K. (2012) Ablation of neurogenesis attenuates recovery of motor function after focal cerebral ischemia in middle-aged mice. PLoS One. 7(10) Kiyoshi CM, Zhou M. (2019) Astrocyte syncytium: a functional reticular system in the brain. Neural Regen Res. 14(4):595-596 Xie M, Yi C, Luo X, Xu S, Yu Z, Tang Y, Zhu W, Du Y, Jia L, Zhang Q, Dong Q, Zhu W, Zhang X, Bu B, Wang W. (2011) Glial gap junctional communication involvement in hippocampal damage after middle cerebral artery occlusion. Ann Neurol. 70(1):121-32

BrdU (50 mg/kg, i.p.) was injected twice daily for five consecutive days, six weeks before perfusion. Thousands of new born cells are generated daily, which rapidly migrate to OB or molecular layer of DG. Therefore, these BrdU-positive cells may not truly reflect the proliferating cells in the SGZ.

It is absolutely right that new nerve cells are constantly being formed and integrated into the existing network. Therefore, by means of BrdU delivery we assessed changes in the number of newly formed neurons at different time points after lesion and compared them with sham animals, thus reflecting changes in neurogenesis due to the early lesion.

A RFP-retroviral vector was injected into the dentate gyrus, which could lead to a trauma, and in turn effect the neurogenesis.

Although we agree with the reviewer about the effects of a possible trauma caused by the intrathecal retroviral injection, sham animals also received a similar injection so that changes observed can only be attributed to the effect of the MCAO. Furthermore, the analysis was done 16 month after injection thereby limiting the possible contribution of this intervention.

The rational may be provided why water maze test was performed in 6 months after stroke.

The aim of this study was to investigate the long-term effects of an ischemic lesion on both neurogenesis and cognitive function. Therefore, we analyzed the number of new neurons as well as the cognitive function after an extended period of 6 months after the lesion.

Reviewer 3 Report

The paper by Kathner-Schaffert and collegues is potentially very interesting. However it is in my view hard to review it in details for the way it is written/communicated.

Below only some suggestions for improvement

-The abstract is not structured and the exp design is spred along the text (e.g on line 4 it shoudl be written when animas are sacrificed instead to get this info on lines 10-11)

-There is no consistency in the way how the exp groups are described in the text (e.g from exp design fig 1, I cannot understand well where the 20 months group is)

-The results section is not written effectively and is not concise as it should be with long list of numerical values and very little text describing the recorded biological effects. I suggest to focus on the description of the most important findings in each figure and provide numerical values and stat only about them. As it is now I cannot follow the flow.

-The figure legends are sometime describing the biological effect while they should be devoted to describe "technically" how the analys was done (there is the main text to describe the results). Importantly the stat used and n=?? should be placed bottom of each legend

Author Response

Reviewer 3

The abstract is not structured and the exp design is spred along the text (e.g on line 4 it shoudl be written when animas are sacrificed instead to get this info on lines 10-11)

The abstract and the description of the experimental design have been improved following the reviewer’s suggestion.

There is no consistency in the way how the exp groups are described in the text (e.g from exp design fig 1, I cannot understand well where the 20 months group is)

We re-arranged the groups and improved the description for better consistency and clarity.

The results section is not written effectively and is not concise as it should be with long list of numerical values and very little text describing the recorded biological effects. I suggest to focus on the description of the most important findings in each figure and provide numerical values and stat only about them. As it is now I cannot follow the flow.

We fully agree with the reviewer’s comment. In the previous version we followed instructions from another reviewer who asked to include all numbers in the text. We know added these numbers into supplementary appendices to improve the clarity and the flow of the text. Only main data were left in the results part.

The figure legends are sometime describing the biological effect while they should be devoted to describe "technically" how the analysis was done (there is the main text to describe the results). Importantly the stat used and n=?? should be placed bottom of each legend

We agree with the reviewer‘s suggestion and changed the figure legends accordingly.

Round 2

Reviewer 3 Report

Ms is improved.

This manuscript is a resubmission of an earlier submission. The following is a list of the peer review reports and author responses from that submission.

Round 1

Reviewer 1 Report

The authors showed that stroke occurring at young age has a long-lasting effect on hippocampal-mediated learning. The data are interesting. However, many descriptions regarding research design and results are not clear. Data are not discussed properly.

Comments:

1.    In the last paragraph of introduction, the authors wrote: “Since the risk of ischemic infarcts increases with age, it is of major clinical interest to investigate the impact of stroke on neurogenesis and cognitive function in the aging brain.” The brain response to stroke injury varies in different age groups. In this study, the stroke was created in young mice, the response might be different from those created in aged mice. The authors should consider revise this sentence.

2.    The experimental design showed in Fig. 1A is difficult to understand. Add a table might be helpful.

3.    The authors should give information about the retroviral vector they have used, such as, what promote have been used in this vector? The authors should give a brief description about the injection procedure, even the procedure has been described in a previous paper. What dose has been injected?

4.    The authors should discuss why the brain volumes were smaller in 9-month-old but not in 20-month-old MCAO mice compared to corresponding controls. It seems that the brain volume of 20-month-old MCAO mice was larger than that of 9-month-old MCAO mice (Fig. 1B).

5.    The authors should also discuss why the infarct volume is smaller in 7.5-month-old mice but not in 9-month-old mice.

6.    In Fig. 3D legend, the authors indicated that the 9-month-old mice showed an impaired spatial learning. The data shows that the spatial learning also impaired in 7.5-month-old mice.

7.    In the first paragraph of page 9, the authors wrote: “After reversal, the main differences were observed in the 6 and 7.5-month-old mice,…” It seems 9-month-old also showed difference.

8.    In the first paragraph of page 9, the authors also wrote: “The 20-month-old stroke group showed significantly more random search paths compared to the controls (Fig. 4). This claim seems not supported by the fig.

9.    In Fig. 5, A label is missing. For Fig 5A picture, the top labels showed be “merged-Nestin-GFP-GFAP/DCX”.

10. Since the overall neurogenesis is reduced in mice subject with MCAO. Have the authors compared the DC volume?

11. The authors indicated that the scale bar in Fig. 1C is 10 mm. Which seems incorrect. Please check.

12. Please indicated scale bar lengths in Figs. 2, 5, and 6.

Author Response

Revision of Manuscript ID: cells-466882 Reviewer: 1 1. In the last paragraph of introduction, the authors wrote: “Since the risk of ischemic infarcts increases with age, it is of major clinical interest to investigate the impact of stroke on neurogenesis and cognitive function in the aging brain.” The brain response to stroke injury varies in different age groups. In this study, the stroke was created in young mice; the response might be different from those created in aged mice. The authors should consider revise this sentence. Response: We agree with the reviewer. We revised the introduction and made changes to the corresponding sentences accordingly.   2. The experimental design showed in Fig. 1A is difficult to understand. Add a table might be helpful.  Response: We revised and improved the figures for a better comprehension of the experimental design. Also, for a better understanding of the data, we present the 20-month group separately from the 6, 7.5 and 9 month-old groups. Further, we changed our graphs into to box blots in order to the distribution of data.  3. The authors should give information about the retroviral vector they have used, such as, what promote have been used in this vector? The authors should give a brief description about the injection procedure; even the procedure has been described in a previous paper. What dose has been injected?  Response: We now describe in detail the viral vectors used and the intrathecal application, as shown below. Page 5, paragraph 2:  “Injection of Retroviral Vector and BrdUA RFP-retroviral vector was injected into the dentate gyrus in the 20-month-old group on day 4 after surgery, in order to assess the impact of stroke on the morphology of newly generated neurons, according to a procedure previously described [5,6]. The CAG-Red Fluorescent Protein (RFP) retroviral vectors were developed from a mouse Moloney leukemia virus by co-transfection of HEK 293 T cells with the compound promoter CAG, the reporter gene RFP, the CMV enhancer protein, the VSV-G rabies virus coating glycoprotein and the Woodchuck Hepatitis Virus post-transcriptional regulatory element (WPRE). The final titer was 1 x 107 colony forming units/ml. For the injections, mice were anesthetized with 2.5 % isoflurane in a N2O : O2 (3 : 1) mixture. A sagittal section was made to open the scalp. The following coordinates were used: lateral -1.5 mm from the midline, -1.9 mm posterior to bregma. A glass cannula, containing 1.2 µl of viral vector was inserted into the opening from the scalp, dorsoventral from the dura mater, 2 mm deep into the brain tissue on the side of the stroke. During injection, body temperature was kept using a heating pad. Sham-operated control mice underwent the same surgical procedure [6].For labeling of proliferating cells, animals were treated with 5-bromo-deoxyuridine (BrdU) (50 mg/kg, i.p.) twice daily for five consecutive days, six weeks before perfusion (Fig. 1A/3A).” 4. The authors should discuss why the brain volumes were smaller in 9-month-old but not in 20-month-old MCAO mice compared to corresponding controls. It seems that the brain volume of 20-month-old MCAO mice was larger than that of 9-month-old MCAO mice (Fig. 1B). The authors should also discuss why the infarct volume is smaller in 7.5-month-old mice but not in 9-month-old mice. Response: We apologize for this discrepancy. We revised our previous data and in addition to the brain- and infarct volume, we now also analyzed the volume of the hippocampus and dentate gyrus in the animals and included the corresponding values in the figures (Fig.1C + D, Fig.3B). After revision and addition of new data we also redo the statistics, and found them not to be normally distributed, using the Shapiro-Wilk test and thus all values were analyzed using the Mann Whitney “U” test. Our new calculations show no significant differences between the groups for all volumes measured. The 9 month-old control group shows a trend to higher global volume and a smaller spread of values compared to the MCAO group. The 20 month-old group shows a relatively stable volume size in both the control as well as the MCAO group. The new statistical analysis also eliminates the significance in the area of infarct volumes. There is a trend towards a greater dispersion in all groups. Globally, the increased spread between MCAO and sham control may be due to stroke-dependent tissue loss in the frontal subcortical regions and differential generation of the glial scar. However, the volumes of the hippocampus and the dentate gyrus are stable and do not differ significantly from each other. 6. In Fig. 3D legend, the authors indicated that the 9-month-old mice showed an impaired spatial learning. The data shows that the spatial learning also impaired in 7.5-month-old mice. Response: We fully agree with the reviewer’s comment. The present study clearly shows strong impairments of learning and re-learning in the 6, 7.5 and 9 month old group. The recall of the platform position previously learned takes place following the learning phase by means of the probe trail. This recall is significantly impaired in the 9 month old group in the first probe trail. The 2nd probe trail shows a reduced learning of the new platform position mainly in the 7.5 and 9 month old group. The MCAO animals show a reduced search in the new target quadrant. To clarify this point we changed the legend of Figure 3D: D Recall of learning was performed using probe trail on day 4 (before reversal) and day 5 (after reversal). At day 4, most deficits in recall were observed in 9 month-old MCAO mice. At day 5 the 7.5 month-old and 9-month-old stroke mice showed an impaired spatial learning compared to sham mice in the probe trail. 7. In the first paragraph of page 9, the authors wrote: “After reversal, the main differences were observed in the 6 and 7.5-month-old mice,…” It seems 9-month-old also showed difference.  Response: We agree with the reviewer. In addition to the deficits in latency and distance in the 6-, 7.5- and 9 month-old MCAO group during learning and re-learning (Fig. 6), we found an increased use of hippocampus-independent search strategies. Following the reviewer’s comment we improved this result part. Page 15, paragraph 2:  Since neurogenesis is strongly associated with hippocampus-dependent learning, we determined the impact of stroke on strategies used by the animals to find the platform (Fig. 7). Using the binary logistic regression analysis we found that all groups used hippocampus-independent strategies in the beginning of the Water Maze, which changed during ongoing training to more hippocampus-dependent strategies. The ratio of hippocampus-dependent/independent strategies (strat) increased during the water maze performances. After the platform location was changed on day 4, fewer hippocampus-dependent strategies were observed, however, these increased again on day 5 (Fig. 7). During the learning phase, the 6, 7.5 and 9-month-old stroke groups used significantly more random search and scanning paths and less direct search paths to locate the platform as compared to sham groups (6 month-group: D1 strat5 p = 0.022; D2 strat2 p = 0.001, strat3 p < 0.001, strat5 p = 0.008; D3 strat 3 p = 0.001, strat5 p = 0.001; 7.5 month-old group: D1 strat 2 p = 0.034; D2 strat 2 p = 0.027; 9 month-old group: D1 strat2 p = 0.042, strat5 p = 0.041; D2 strat2 p = 0.005). After reversal, all MCAO groups used significantly less hippocampus-dependent strategies on day 5 (6 month-old group: strat3 p< 0.001, strat7 p = 0.022; 7.5 month-old group: strat2 p = 0.016, strat3 p = 0.019, strat5 p = 0.012, strat6 p = 0.036, strat7 p = 0.005; 9 month-old group: strat2 p = 0.029). In summary, we observed significant changes in the use of hippocampus-dependent strategies between MCAO und sham groups (hippocampal-dependent strategies: sham versus MCAO: 6-month-group: D1 p = 0.030; D2 p = 0.001; D3 p = 0.042; D5 p = 0.042; 7.5-month-group: D5 p = 0.004; 9-month-old group: D5 p = 0.02) (Fig. 7B-D).The 20-month-old stroke group showed significantly more random search paths (p = 0.029) compared to the controls (Fig. 7).Stroke mice showed strong long-term deficits in learning and re-learning in the Morris water maze, both in terms of classical parameters and hippocampal-dependent strategies. 8. In the first paragraph of page 9, the authors also wrote: “The 20-month-old stroke group showed significantly more random search paths compared to the controls (Fig. 4). This claim seems not supported by the fig.  Response: We again carefully checked all the statistical data. For the strategies we used the matlab script which has already used and published by Garthe et al. (2009). For the statistical analysis of the different search strategies we used the binary logistic regression analysis and statistical differences were found using the generalized estimating equations method (GENLIN Procedure) which was already published (Woitke et al. (2017); Urbach et al. (2017). The 20 month-old group showed significantly more random searches (p = 0.029) indicating increased use of hippocampal-independent strategies after MCAO. The statistical differences represent the complete day. 9. In Fig. 5, A label is missing. For Fig 5A picture, the top labels showed be “merged-Nestin-GFP-GFAP/DCX”. Response: We improved the label of the figure accordingly.  10. Since the overall neurogenesis is reduced in mice subject with MCAO. Have the authors compared the DC volume? Response: We measured the DG volumes and could not find any significant differences between the groups. The DG volumes are now included in figures 1 and 3. 11. The authors indicated that the scale bar in Fig. 1C is 10 mm. Which seems incorrect. Please check. Response: We apologize for this error. We changed the scale bar to 1 mm. 12. Please indicated scale bar lengths in Figs. 2, 5, and 6. Response: We now included the scale bars as indicated.

Reviewer 2 Report

In the present study, Kathner and colleagues evaluated the impact of stroke on adult neurogenesis and hippocampally-mediated cognitive decline. The authors induced a stroke in mice at three months of age using the middle cerebral artery occlusion model. Following surgery, mice were injected with a RFP-retroviral vector to assess the impact of stroke on the morphology of newly generated neurons. At 6, 7.5, and 9 months, the mice were tested on the Morris water maze to assess the effect of stroke and age on hippocampally-dependent learning. The authors claim to find decreased neurogenesis and decreased cognitive performance in stroke mice.

I found this study both insufficiently motivated and lacking in adequate statistical procedures. First, from the introduction, it was unclear both what is currently known and what the hypothesis and predictions of the present study were. Based on the text on page 2, it would appear that all ischemic strokes impact the hippocampus. Is that true? The authors likewise state that ischemic lesions increase the number of newborn neurons. On page 12, the authors claim to have found a "significant stroke-dependent reduction of adult neurogenesis." Is that finding not counter to the known literature described in the introduction? Because the authors do not provide a clear hypothesis and predictions, it is unclear why they chose the present experimental design. Why was the stroke induced at the same age across mice, rather than at different ages? Additionally, the authors refer the reader to another publication to find the meaning behind the "search strategies" used and whether those strategies rely on the hippocampus. These appear critical to the claims the authors make and thus should be described in the present manuscript. If the stroke procedure is supposed to increase hippocampal neurogenesis (though this itself is unclear), then wouldn't one predict greater dependence on hippocampal strategies, and not less? Why weren't the animals tested on the morris water maze before stroke? 

Second, despite a brief mention of statistical analysis on page 5, there is virtually no information provided regarding the statistical tests performed in this paper. Many results are described qualitatively without quantitative support. Furthermore significance is merely indicated by the authors stating that there was a significant effect. Without accompanying statistical tests and values (e.g. t-statistics, degrees of freedom, and p-values), it is impossible to evaluate the results of the manuscript. Additionally, there is no indication that the authors used appropriate correction for multiple comparisons. They state that "P values < 0.05 were considered statistically significant" (page 5), but that would be inappropriate for a number of the analyses, including several shown in Figure 3 where performance on the water maze was compared over several days in the same animals. 

Author Response

Revision of Manuscript ID: cells-466882

 Reviewer 2 

First, from the introduction, it was unclear both what is currently known and what the hypothesis and predictions of the present study were.

 Response: We fully agree with the reviewer’s comment and improved the introduction to highlight the hypothesis and aim of the study. 

Based on the text on page 2, it would appear that all ischemic strokes impact the hippocampus. Is that true?

Response: The MCAO mainly affects the frontal subcortical areas as shown in Figure 1E, with perilesional areas extending to the hippocampus. However, the hippocampus and dentate gyrus per se are not affected. To further clarify this aspect, we additionally determined volumes of the hippocampus and dentate gyrus, which showed no significant differences. We added new figures (Fig.1C + D, Fig.3B). Due to the new volume data we revised the statistics. First, we re-checked the data distribution using the Shapiro-Wilk test. Since data are not normally distributed we analyzed all volume data using the Mann Whitney “U” test. The new analysis shows no significant differences between the groups for all volumes measured. Further, we used box plots to show the dispersion in all groups. The volumes of the hippocampus and the dentate gyrus are stable and do not differ significantly from each other.  

The authors likewise state that ischemic lesions increase the number of newborn neurons. On page 12, the authors claim to have found a "significant stroke-dependent reduction of adult neurogenesis." Is that finding not counter to the known literature described in the introduction? Because the authors do not provide a clear hypothesis and predictions, it is unclear why they chose the present experimental design. Why was the stroke induced at the same age across mice, rather than at different ages?

 Response: We improved the introduction to clarify stroke-dependent in changes neurogenesis, which has already been described.  Page 3, paragraph 1:

Most studies evaluate the short-term impact of stroke on the brain [10-12]; however, to what extent a stroke lesion early in life affects neural precursor populations, neurogenesis and integration of new neurons over a long period of time after lesion is still not fully understood. Furthermore, whether alterations in the neurogenic niche are associated with changes in brain function (i.e. learning and memory) has yet to be addressed. In our previous studies we showed, that early stroke induced in young adult mice resulted in a faster maturation of immature neurons and thereby generated a portion of aberrant neurons. Therefore, here we evaluated the impact of a prefrontal stroke induced in the young mice on the neurogenic niche and cognitive function in mice during aging. We hypothesized that a stroke lesion in young mice, which significantly increases neurogenesis during the first weeks, would disturb the neurogenic niche, and lead to a long-lasting cognitive impairment during aging. To test our hypothesis, we induced a stroke in 3-month-old mice using the MCAO model and evaluated the cellular and functional consequences in the dentate gyrus during aging. To assess long-term effects on cognitive outcome we used a modified version of the Morris Water Maze which permits to use of a re-learning paradigm and the differentiation of hippocampus dependent- and independent search strategies.

Additionally, the authors refer the reader to another publication to find the meaning behind the "search strategies" used and whether those strategies rely on the hippocampus. These appear critical to the claims the authors make and thus should be described in the present manuscript. If the stroke procedure is supposed to increase hippocampal neurogenesis (though this itself is unclear), then wouldn't one predict greater dependence on hippocampal strategies, and not less?

Response: We agree with the reviewer and accordingly included the following text:

 Page 18, paragraph 2:

“The contribution of post-stroke neurogenesis on cognitive function is still under debate and not clear so far. Under physiological conditions, a decrease of neurogenesis due to aging or different treatments (cytostatic drugs, knockout model) correlates with an impairment in cognitive function, represented by a reduction of hippocampal-dependent strategies. Consequently, an impairment in neurogenesis leads to a reduced use of hippocampal-dependent strategies. Under pathophysiological conditions such as stroke, it is not clear to what extent network activity is disturbed and in which way these changes are due to the reduction of neurogenesis. Directly after stroke, neurogenesis increases, which correlates with learning and memory deficits and further with the reduction of hippocampal-dependent strategies in the water maze. Underlying mechanisms are not clear, but may involve neuronal network disturbance caused by the presence of aberrant neurons (already described in young mice after stroke) which lead to wrong connectivity. In the long run, these false network connectivity`s may also lead to decreased cognition as reflected by a lesser use of hippocampal-dependent strategies. It is also possible that strongly reduced neurogenesis over a long term per se leads to impairment of cognition. The possible mechanisms linking changes in neurogenesis and network connectivity to learning and memory are still not clear and need further investigations”.

Why weren't the animals tested on the morris water maze before stroke? 

Response: Usually there is no baseline definition for the water maze test, since the test itself has a training effect on the animals.  Since the purpose of our study was to find a possible correlation between changes in neurogenesis, which takes several weeks, and associated cognitive impairment after stroke, a pre-lesion test strategy would have focused mainly on memory and would have required a cross-sectional study and not a longitudinal one, due to the training effect ot the test. However, this approach would be interesting to further characterize the impact of stroke on cognitive processes.

Second, despite a brief mention of statistical analysis on page 5, there is virtually no information provided regarding the statistical tests performed in this paper. Many results are described qualitatively without quantitative support. Furthermore significance is merely indicated by the authors stating that there was a significant effect. Without accompanying statistical tests and values (e.g. t-statistics, degrees of freedom, and p-values), it is impossible to evaluate the results of the manuscript. Additionally, there is no indication that the authors used appropriate correction for multiple comparisons. They state that "P values < 0.05 were considered statistically significant" (page 5), but that would be inappropriate for a number of the analyses, including several shown in Figure 3 where performance on the water maze was compared over several days in the same animals. 

Response: We improved the manuscript following the reviewer’s suggestions: we changed the columns to box plots to make the distribution of the data more visible. Further, we described our statistical tests in detail in the paragraph: Quantification and Statistical Analysis (page 9). We first used the Shapiro-Wilk test to verify the data. If data were not normally distributed, we applied the Mann-Whitney “U” test, as it is the case for all cell quantifications, volume measurements and probe trails of the water maze. For latency and distance calculations we used the two-way repeated measure ANOVA followed by Bonferroni post hoc test, to verify the significant differences between the groups. For the analyses at different time points within the groups we used the t-test. In order to verify the differences in search strategies we used the generalized estimating equations method (GENLIN Procedure), which was previously used by Woitke et al. 2017 and Urbach et al. 2017. Our data are reported as the median and were considered statistically significant for p values < 0.05. Classical parameters of the Water Maze (latency and distance curves) are given as mean ± SEM (standard error of the mean). We added the statistical procedures used in the corresponding legends.

Round 2

Reviewer 1 Report

The authors addressed most of my questions, but not all.

Comments:

I asked on my last review that the authors should discuss why the brain volumes were smaller in 9-month-old but not in 20-month-old MCAO mice compared to corresponding controls. It seems that the brain volume of 20-month-old MCAO mice was larger than that of 9-month-old MCAO mice (Fig. 1B). The authors did not answer this question, instead they remove 20-month-old group from Fig. 1B.

The location of Fig. 2 and 3 are not in order. It is confusing.

Author Response

I asked on my last review that the authors should discuss why the brain volumes were smaller in 9-month-old but not in 20-month-old MCAO mice compared to corresponding controls. It seems that the brain volume of 20-month-old MCAO mice was larger than that of 9-month-old MCAO mice (Fig. 1B). The authors did not answer this question, instead they remove 20-month-old group from Fig. 1B.

Response:

We compared changes in brain volume between the MCAO groups and between sham and MCAO groups at the different ages. The analysis was carried on using the Mann-Whitney-U-test and we did not detect any significant differences. Also, no significant differences were detected between 9 and 20 month-old MCAO groups. The 9 month old sham group showed a significant higher brain volume compared to the 20 month-old sham group. The brain volume of the 20-month-old MCAO mice is not larger than that of the 9 month-old mice. We added all statistical data into the manuscript. The 20 month old group was included in Fig. 3.

The location of Fig. 2 and 3 are not in order. It is confusing.

Response:

The arrangement of the figures in the manuscript was carried out by the Editorial Office. We will alert the editor for the proper placement of the figures in the text.

Reviewer 2 Report

I appreciate the effort that the authors' took to address my comments; however, substantial critical information is still missing from the manuscript and my concerns remain. 

1. Statistical tests are still not reported properly and correction for multiple comparisons does not appear to have been applied correctly.

For Mann-Whitney U tests and t-tests, no test statistics are reported. When reporting a Mann-Whitney U test, the value of U must be reported along with the sample size(s), e.g. U = 10, N = 8, p = 0.03. When reporting a t-test, the t-statistic and degrees of freedom must be reported, e.g. t(8) = 10. 

The authors present the results of at least one 2-way ANOVA (page 12; it is unclear whether there is a single or multiple ANOVAs); however the factors and results are not adequately described. As there are three possible factors with multiple levels (behavioral performance: distance, latency; group: MCAO, sham; age: 6, 7.5, 9, 20 month), it is impossible to decipher on what data the ANOVAs were conducted. Further, all reports of F-statistics should be accompanied by degrees of freedom.

There remain a number of instances where the authors claim no effects, but do not report any statistics. e.g. "No conclusions could be drawn about the physical performance of the mice based on the swimming speed" (page 11). First, it's unclear why the authors interchange "velocity" and "swimming speed" if both mean the same thing. Second, the authors refer to the figure to support this statement, but Figure 6 does not include any velocity data. Another example is on page 12, where the authors state "no statistical differences were observed in the 20 month-old group." Again, the test statistics, degrees of freedom and p-values should be reported. Visual inspection of Figure 6 is insufficient to support such a claim. There are further statements not supported by statistical tests throughout page 12.

The authors state that they applied Bonferroni correction, but the post-hoc tests comparing each MCAO group to the control are not Bonferroni corrected. For 6-month-old mice, five comparisons were made, one for each day making the Bonferroni corrected p-value = 0.01. Therefore, of the seven reportedly significant effects, only latency on Day 2 and distance on Days 2 and 3 would remain significant. The same applies to any comparisons made with the 7.5-month-old, 9-month-old, and 20-month-old groups.

2. Hippocampal-dependent strategies

The modification of Figure 7 has helped to clarify hippocampal dependent vs. independent strategies; however, information is still missing. How were strategies coded or operationalized? The text states only that "Analyses of the different search strategies were performed using the Matlab software." Presumably some algorithms are required to classify paths as "direct swim," "focal search," etc. How was that done? Percentages of each path type should be reported in the manuscript, e.g. "During the learning phase, the 6, 7.5 and 9-month-old stroke groups used significantly more random search and scanning paths and less direct search paths to locate the platform as compared to sham groups." As above, this also requires Bonferroni correction for multiple comparisons. 

3. Conclusions made

The authors claim, "Our results clearly demonstrate a stroke-dependent reduction of dentate neurogenesis correlating with deficits in flexible learning and a decline of the usage of hippocampus-dependent strategies, indicating memory impairment."

The authors show (though without appropriate test statistics) that there is a reduction in proliferation cells and survival of new neurons as mice age (Figure 2). The biggest behavioral impairments, increases in distance and latency to platform, are found for 6 month, not 9 month mice (Figure 7). Further, strategy differences are also clearest between 6 month MCAO and sham mice. The results are difficult to interpret without accompanying statistical information, but, if anything, the results show nearly the opposite of what the authors claim. The biggest decrement in performance is seen for mice with the least reduction in proliferation. 

The authors state, "It was previously shown that neurogenesis decreases between 9 and 20-month-old mice, which was also observed in our study." Presumably, then, the 20-month-old mice should show the biggest behavioral impairments, and yet of all the groups, 20-month-old MCAO mice do not differ from same-aged sham mice. 

Author Response

I appreciate the effort that the authors' took to address my comments; however, substantial critical information is still missing from the manuscript and my concerns remain.

1. Statistical tests are still not reported properly and correction for multiple comparisons does not appear to have been applied correctly.

For Mann-Whitney U tests and t-tests, no test statistics are reported. When reporting a Mann-Whitney U test, the value of U must be reported along with the sample size(s), e.g. U = 10, N = 8, p = 0.03. When reporting a t-test, the t-statistic and degrees of freedom must be reported, e.g. t(8) = 10.

The authors present the results of at least one 2-way ANOVA (page 12; it is unclear whether there is a single or multiple ANOVAs); however the factors and results are not adequately described. As there are three possible factors with multiple levels (behavioral performance: distance, latency; group: MCAO, sham; age: 6, 7.5, 9, 20 month), it is impossible to decipher on what data the ANOVAs were conducted. Further, all reports of F-statistics should be accompanied by degrees of freedom.

Response:

We further revised the statistical analyses with support of our Institute for Medical Statistics, Computer Science and Data Science. As already mentioned, we performed our analyses based on current data and own previous publications (e.g Urbach et al. 2017, Woitke et al. 2017). As indicated by the reviewer, we now included detailed description of the statistical tools used in our analyses in the ‘Methods’ section:

Page 10, paragraph 2:

“Statistical Analysis

Statistical analyses were performed using SPSS 22.0 for Windows (IBM Corp., Armonk NY). Data were tested for normality using the Shapiro-Wilk test. Data which failed were analysed using the Mann-Whitney “U” test. All cell quantifications, volumetry data and probe trail of the Water Maze were calculated by Mann-Whitney “U” test. The data are reported as the median (Mdn) and were considered statistically significant for p values < 0.05. 

Sholl analysis of dendritic complexity in the 20 month old group was tested using the one-way ANOVA (dependent variable: intersections, factor: groups (sham versus stroke). The data are reported as mean ± SEM and were considered statistically significant for p values < 0.05.

For classic parameters (latency, distance and velocity) and hippocampus-dependent strategies of the Water Maze, statistical analyses were performed as follows:

(1) For the analyses of MCAO and sham control groups at the different ages, a 2 way-ANOVA with repeated measures and post-hoc Tukey test was performed (inner-subject variables: days and trails; between subject factor: ages).

(2) For the analyses between the groups MCAO versus sham controls at the different ages, a 2 way-ANOVA with repeated measures and post-hoc Tukey test was used (inner-subject variables: days and trails; between subject factor: groups).

(3) The comparison per day (MCAO versus sham controls) at the different ages, a fractional-ANOVA with post-hoc Tukey test was performed (dependent variables: days; factor: groups).

(4) For the analysis of hippocampus-dependent strategies for both groups (MCAO, sham controls) at the different ages, a 2 way-ANOVA with repeated measures and post-hoc Tukey test was performed (inner-subject variables: days and strategies; between subject factor: ages).

(5) For the analysis of hippocampus-dependent strategies between the groups (MCAO versus sham controls) at the different ages 2 way-ANOVA with repeated measures and post-hoc Tukey test was performed (inner-subject variables: days and strategies; between subject factor: groups).

(6) The comparison per day (MCAO versus sham controls) at the different ages was performed using a fractional-ANOVA with post-hoc Tukey test (dependent variables: days; factor: groups).

Classical parameters of the Water Maze (latency, distance and velocity) and hippocampus-dependent strategies are given as mean ± SEM

Search strategies used in the Water Maze were analysed using an algorithm based on the generalized estimating equations method (GENLIN Procedure) [14].”

We have listed all the F-values accordingly and added the degrees of freedom.

There remain a number of instances where the authors claim no effects, but do not report any statistics. e.g. "No conclusions could be drawn about the physical performance of the mice based on the swimming speed" (page 11). First, it's unclear why the authors interchange "velocity" and "swimming speed" if both mean the same thing. Second, the authors refer to the figure to support this statement, but Figure 6 does not include any velocity data. Another example is on page 12, where the authors state "no statistical differences were observed in the 20 month-old group." Again, the test statistics, degrees of freedom and p-values should be reported. Visual inspection of Figure 6 is insufficient to support such a claim. There are further statements not supported by statistical tests throughout page 12.

We agree with the reviewer and now use the term velocity for the physical performance. In addition, we added the velocity diagrams to figure 6 and completed the manuscript with all statistical comparisons for velocity (page 21), latency, distance and sholl analysis. We added a supplemental table for statistical data of the sholl analysis.

The authors state that they applied Bonferroni correction, but the post-hoc tests comparing each MCAO group to the control are not Bonferroni corrected. For 6-month-old mice, five comparisons were made, one for each day making the Bonferroni corrected p-value = 0.01. Therefore, of the seven reportedly significant effects, only latency on Day 2 and distance on Days 2 and 3 would remain significant. The same applies to any comparisons made with the 7.5-month-old, 9-month-old, and 20-month-old groups.

We revised the statistical data and expanded the description of the statistical tests used in the Methods section, as described above.

2. Hippocampal-dependent strategies

The modification of Figure 7 has helped to clarify hippocampal dependent vs. independent strategies; however, information is still missing. How were strategies coded or operationalized? The text states only that "Analyses of the different search strategies were performed using the Matlab software." Presumably some algorithms are required to classify paths as "direct swim," "focal search," etc. How was that done? Percentages of each path type should be reported in the manuscript, e.g. "During the learning phase, the 6, 7.5 and 9-month-old stroke groups used significantly more random search and scanning paths and less direct search paths to locate the platform as compared to sham groups." As above, this also requires Bonferroni correction for multiple comparisons.

We added the explanation of the different search strategies for a better understanding.

The analysis of the different search strategies was performed with the algorithm based on the generalized estimating equations method (GENLIN Procedure). The statistical method is already published by Woitke et al. (2017) and Urbach et al. 2017.

3. Conclusions made

The authors claim, "Our results clearly demonstrate a stroke-dependent reduction of dentate neurogenesis correlating with deficits in flexible learning and a decline of the usage of hippocampus-dependent strategies, indicating memory impairment."

The authors show (though without appropriate test statistics) that there is a reduction in proliferation cells and survival of new neurons as mice age (Figure 2). The biggest behavioral impairments, increases in distance and latency to platform, are found for 6 month, not 9 month mice (Figure 7). Further, strategy differences are also clearest between 6 month MCAO and sham mice. The results are difficult to interpret without accompanying statistical information, but, if anything, the results show nearly the opposite of what the authors claim. The biggest decrement in performance is seen for mice with the least reduction in proliferation.

The authors state, "It was previously shown that neurogenesis decreases between 9 and 20-month-old mice, which was also observed in our study." Presumably, then, the 20-month-old mice should show the biggest behavioral impairments, and yet of all the groups, 20-month-old MCAO mice do not differ from same-aged sham mice.

After a stroke, neurogenesis increases rapidly within a very short time and drops sharply during aging up to the age of 20 months. The present study shows a significant stroke-dependent reduction of neurogenesis between 6 and 9 months and between 9 and 20 months. The data also show that neurogenesis decreases more rapidly after stroke between 7.5 and 9 months and slightly up to 20 months, whereby the MCAO group reaches the sham level. Regarding learning and re-learning, most changes are present in 6 and 7.5 month groups but also the memory performance (probe trail) in 7.5 and 9 month-old mice is disturbed after reversal. Further the quality of learning using different search strategies is changed in all MCAO groups compared to controls shifting to more hippocampus-independent strategies. Our results indicate a long-term decline in neurogenesis, which is accompanied by learning and memory deficits. The higher drop in neurogenesis in the first 9 month was associated with higher learning deficits compared to the aged mice. In the aged mice the level of neurogenesis, in MCAO and sham controls, was similar and there were no obvious differences in learning impairment between MCAO and sham controls as compared to the middle aged mice.  But beside changes in neurogenesis we found slight differences in the number of immature neurons between MCAO and sham controls, which was accompanied by changes in search strategies. Our current data are relevant for the field of neurogenesis per se and also for the elucidation of mechanisms involved in long-term impairment of brain function after a brain lesion.